# Global forest fragmentation change from 2000 to 2020

Jun Ma [1,2] ✉, Jiawei Li [1,2], Wanben Wu [1] & Jiajia Liu [1] ✉

A comprehensive quantification of global forest fragmentation is urgently required to guide forest protection, restoration and reforestation policies. Previous efforts focused on the static distribution patterns of forest remnants, potentially neglecting dynamic changes in forest landscapes. Here, we map global distribution of forest fragments and their temporal changes between 2000 and 2020. We find that forest landscapes in the tropics were relatively intact, yet these areas experienced the most severe fragmentation over the past two decades. In contrast, 75.1% of the world's forests experienced a decrease in fragmentation, and forest fragmentation in most fragmented temperate and subtropical regions, mainly in northern Eurasia and South China, declined between 2000 and 2020. We also identify eight modes of fragmentation that indicate different recovery or degradation states. Our findings underscore the need to curb deforestation and increase connectivity among forest fragments, especially in tropical areas.

Forest fragmentation is a major driver of global biodiversity loss and ecosystem degradation[1-4]. Identifying which areas have the most severe forest fragmentation is a fundamental task in ecology, but the findings from investigations into the patterns of global forest fragmentation have been inconsistent[5-7]. Forests in tropical regions are regarded as more contiguous than those in other regions[5,8] and Morreale et al.[9] concluded that temperate forests are 1.5 times more fragmented than tropical forests. In contrast, other studies implied that tropical forests are undergoing the most severe forest fragmentation because of an acceleration in deforestation in these regions[10-12]. For example, the fraction of forest edge area to total forest area in the tropics increased from 27% in 2000 to 31% in 2010[12], and there was a net loss of forest cover in the tropics in the same period, while many countries in temperate regions achieved net forest gains[13,14]. These contrasting findings may have been due to a variation in how forest fragmentation was defined on different temporal scales. To gain a better understanding of the consequences of forest fragmentation, a comprehensive quantification of global forest fragmentation is urgently needed.

Forest fragmentation is a landscape-level process of forest change, and mainly occurs over time as decreased patch size, increased patch number of patches, and more forest edges[15]. Accurate assessments and maps of forest fragmentation are crucial for exploring its effects on biodiversity and ecosystem functions. However, previous studies on global forest fragmentation mapping have relied on evaluating static landscape patterns[5,6,8] and have ignored the fact that forest fragmentation is a dynamic process over time. Moreover, although the effects of forest fragmentation can persist for up to a century[16], the effects are mostly immediate and obvious for only the first several decades after formation[17,18]. Therefore, accurate quantification of the current patterns and dynamics of forest fragmentation is key to preventing future biodiversity loss and ecosystem degradation. However, the lack of quantification of global forest fragmentation over recent decades from a dynamic perspective, represents an important knowledge gap.

Over the past several decades, the assessment of forest fragmentation has become increasingly complex because deforestation and afforestation is highly dynamic. Forest fragmentation is generally thought to be associated with forest loss[5,15], and it is thus generally assessed in terms of deforested area. As such, most previous studies on forest fragmentation dynamics focused on tropical forests with high deforestation rates[6,11,12]. However, forest fragmentation and forest cover changes are relatively independent and do not always proceed in

[1]Ministry of Education Key Laboratory for Biodiversity Science and Ecological Engineering, Coastal Ecosystems Research Station of the Yangtze River Estuary, Institute of Biodiversty Science, School of Life Sciences, Fudan University, #2005 Songhu Road, Shanghai 200438, China. [2]These authors contributed equally: Jun Ma, Jiawei Li. ✉e-mail: ma_jun@fudan.edu.cn; liujiajia@fudan.edu.cn

the same direction[7]. Many temperate and subtropical countries have gained forest cover[19], yet forest cover gains do not reverse the trend of forest fragmentation[7]. Consequently, the credibility of large-scale estimates of forest landscape pattern dynamics will be reduced when considering only either fragmentation or coverage. Thus, there is an urgent need to effectively integrate and analyze changes in forest cover and fragmentation patterns to inform forest management decisions.

Here, we constructed a synthetic forest fragmentation index (FFI) to represent the main characteristics of forest fragmentation, including edge, isolation, and patch size effects. We used the FFIs in 2000 and 2020 and their differences (ΔFFI) to determine the static and dynamic patterns of global forest fragmentation. These two indexes were compared among various regions globally to explore which index more effectively and accurately reflects the fragmentation status. We also analyzed key processes and potential causes for some global hotspots of forest fragmentation, and combined changes in forest coverage (ΔFC) and fragmentation to derive a two-dimensional framework for the assessment of forest landscape dynamic patterns. Our study reveals that 75.1% the world's forest landscapes experienced decreased fragmentation but those in tropical regions experienced increased fragmentation during the first two decades of the 21st century.

## Results

### Differences in static and dynamic forest fragmentation index values

We calculated the static FFI using the average weighted values of normalized edge density (ED), patch density (PD), and mean patch area (MPA, using 1-normalized MPA) for 2000 and 2020 (Fig. 1), while the values for the individual normalized ED, PD, and MPA layers are shown in Supplementary Fig. 1. Forest landscapes with low static fragmentation (FFI < 0.2) were mainly in the tropics, western Canada, western Siberia, and Far East Russia, while forests with high static fragmentation (FFI > 0.8) were mainly in eastern North America, southern Europe, central and South China, and along the edges of tropical forests.

The area proportion of forest landscapes with different fragmentation levels remained stable during 2000–2020. There was a slight increase in the percentage of forest landscapes with low static fragmentation from 2000 to 2020 (17% to 19%), and the percentage of forest landscapes with high static fragmentation also decreased slightly from 17% in 2000 to 13% in 2020.

However, the dynamic FFI (ΔFFI) for the period from 2000 to 2020 exhibited a dramatically different pattern from the static FFIs. Approximately 75.1% of global forest landscapes showed a decline in fragmentation (ΔFFI < 0) especially in the western Canada, western and the Far East Russia, and central and South China (Fig. 1c). Forest landscapes with increased fragmentation trends (ΔFFI > 0) were mainly in tropical areas, especially the southeastern Amazon, the Congo Basin, Indochina Peninsula, and some regions in western North America and central Siberia. Meanwhile, fragmentation was relatively stable (ΔFFI ~ 0) in the central Amazon, central and eastern Europe, and the southeastern US.

We then compared the static FFI and ΔFFI across climate zones (Fig. 1d) and found that the world's most fragmented forests were distributed in the subtropics, while the most intact forests were in the tropics and boreal regions. Forests in the subtropical zone had significantly higher static FFI values than those in the other zones in both years ($0.64 \pm 0.34$ in 2000 and $0.62 \pm 0.31$ in 2020, $P < 0.001$), while forests in the tropical regions ($0.43 \pm 0.38$, $P < 0.001$) and boreal regions ($0.41 \pm 0.20$, $P < 0.001$) had the lowest static FFI values in 2000 and 2020, respectively. However, the mean ΔFFI value in the tropics ($0.01 \pm 0.104$, $P < 0.001$) was significantly higher than in the other zones (−0.06 to −0.02). In addition, the ΔFFI generally had a significant negative relationship with the $FFI_{2000}$ across all climatic zones (using a generalized additive model, $P < 0.001$ for all four zones; Supplementary Fig. 2). Furthermore, the static FFIs were significantly and positively correlated with altitude, while there was a significantly negative correlation between ΔFFI and altitude (Supplementary Fig. 3), indicating lowland forests were relatively intact but experienced more severe fragmentation during 2000–2020.

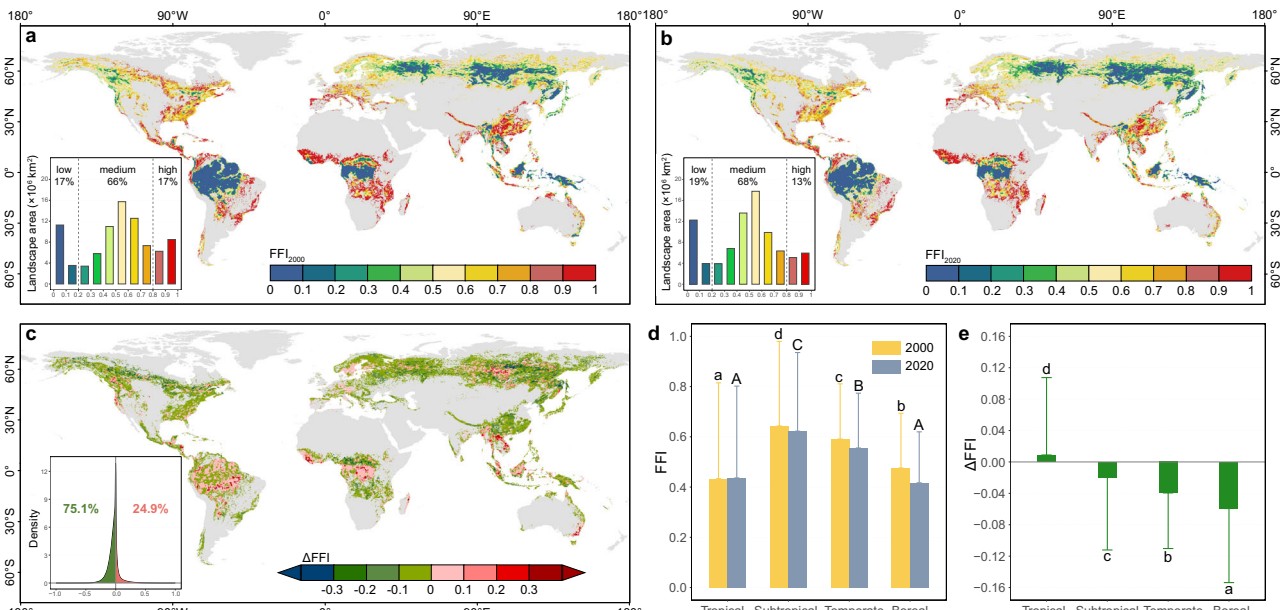

**Fig. 1 | Global distributions of the static forest fragmentation index (FFI) and the dynamic FFI (ΔFFI) for global forest landscapes. a** Static FFI in 2000, (**b**) static FFI in 2020, (**c**) ΔFFI from 2000 to 2020; and comparisons of (**d**) FFI values for 2000 and 2020, and (**e**) ΔFFI values across climatic zones. **d**, **e** The bar heights showed mean value, and error bars showed one standard deviation. The significance of the differences in static FFIs and ΔFFI among climatic zones was tested using the two-side Tukey-HSD test, which adjusted for multiple comparisons, and letters in each bar showed post-hoc differences in mean static FFIs and ΔFFI with $P < 0.001$. **d** The numbers of forest pixels (*n*) from tropical to boreal zones are 581,649, 381,768, 1,031,907, and 1,384,718 in 2000, and 569,260, 378,518, 1,055,401, and 1,385,159 in 2020, respectively. **e** The numbers of forest pixels (*n*) from tropical to boreal zones are 553,655, 363,858, 1,021,172, and 1,320,564, respectively.

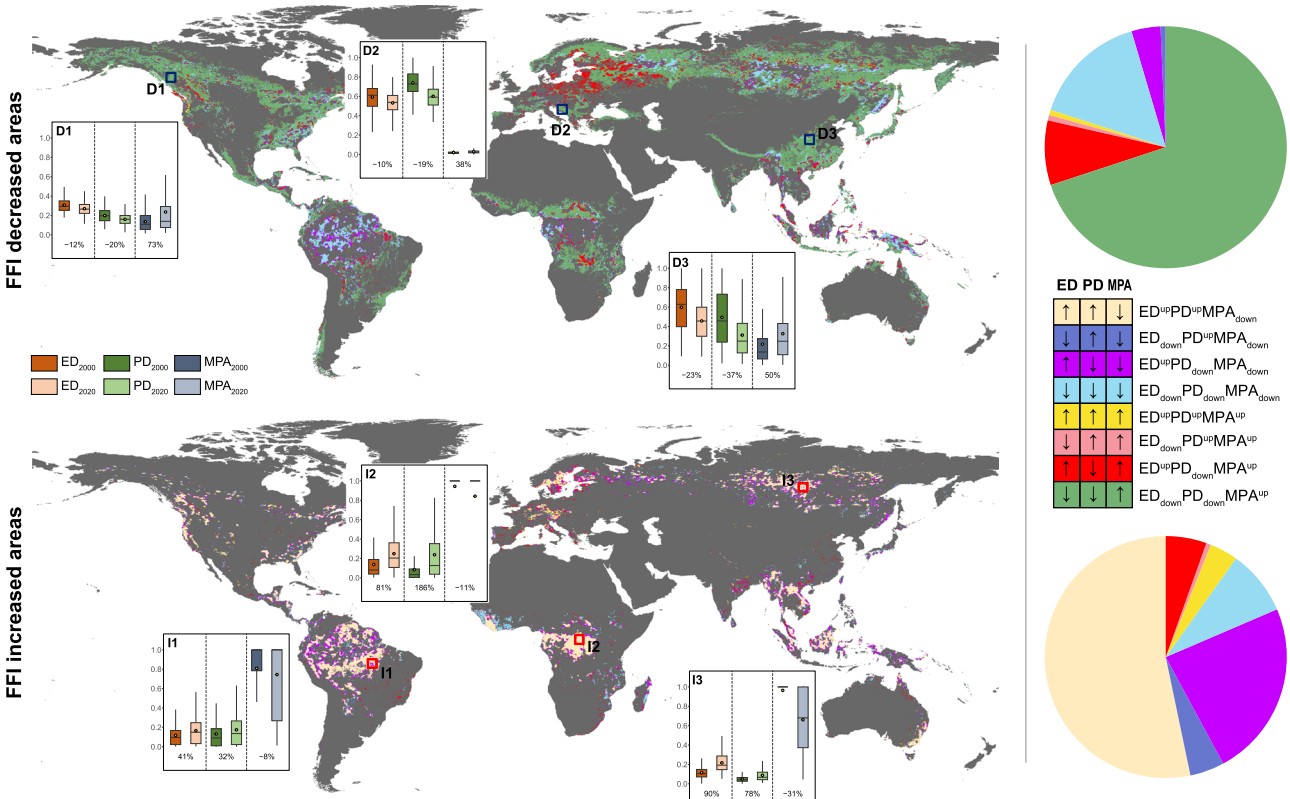

**Fig. 2 | Spatial distributions and composition proportions of eight forest fragmentation process modes for forest fragmentation decreased areas (the upper row) and forest fragmentation increased areas (the below row).** ED, PD, and MPA mean the individual components of the synthetic forest fragmentation index (FFI). The marks of "up" and "down" after each FFI component represent an increase and decrease trend during 2000–2020, respectively. The pie charts on the right represent the area proportions of eight forest fragmentation process modes in FFI decreased (the number of forest pixels, $n = 2,470,511$) and increased areas ($n = 820,937$). Three hotspots in the most obvious FFI decreased and increased areas were selected respectively to evaluate the changes in each component of FFI. Hotspots D1-D3 were in western Canada ($n = 7007$), southern Europe ($n = 3610$), and central China ($n = 5180$), and hotspots I1-I3 were in the southeastern Amazon ($n = 2710$), the Congo Basin ($n = 5102$), and central Siberia ($n = 6640$), respectively. The boxes of hotspots display the median value, lower 25% and upper 75% quartiles; the dots represent the mean value; and the whiskers are extended to the limit of the 1.5-fold interquartile ranges (IQRs).

## Modes and potential causes of forest fragmentation processes

We identified eight modes of forest fragmentation based on the possible combinations of change (increase or decrease) in the three individual components of FFI (ED, PD, and MPA) and detected the area composition proportions of the eight modes for areas with decreased ($\Delta$FFI < 0) and increased ($\Delta$FFI > 0) forest fragmentation, respectively. Among the areas with decreased FFI, the $ED_{down}PD_{down}MPA^{up}$, with an area proportion of 69.8%, was the most common mode and widely distributed all over the world (Fig. 2). Other modes, such as $ED_{down}PD_{down}MPA_{down}$ and $ED^{up}PD_{down}MPA^{up}$ had area proportions of 15.4% and 8.6% and were mainly dominant in the central Amazon and eastern Europe, respectively. In terms of the areas with increased FFI, the $ED^{up}PD^{up}MPA_{down}$ was the most common mode and accounted for 53.3% of the total, which occurred mainly in the tropics, western North America, northern Europe, and central Siberia. The $ED^{up}PD_{down}MPA_{down}$ and $ED_{down}PD_{down}MPA_{down}$ modes had area proportions of 23.6% and 8.7%, respectively, and were predominant in the tropics, Russia, and western Africa.

We also detected the changes in individual fragmentation-related metrics for hotspots in areas with decreased and increased FFI, respectively. Overall, hotspots where FFI decreased had declines in ED and PD, and increases in MPA. Specifically, MPA increased significantly by 73% in western Canada, by 38% in southern Europe, and by 50% in central China from 2000 to 2020 (Fig. 2). Conversely, increased ED and PD played a more important role in hotspots where FFI increased. ED increased dramatically by 41% in the southeastern Amazon, by 81% in

the Congo Basin, and by 90% in central Siberia, while the increments of PD in these hotspots were 32%, 186%, and 78%, respectively. However, MPA decreased slightly (−8% to −31%) in these hotspots where forest fragmentation increased.

Using generalized linear models, we further explored the relationships between $\Delta$FFI and explanatory factors (see Methods) for the globe and the six hotspots. Although $\Delta$FFI was not significantly correlated with any explanatory variables at the global scale (Supplementary Fig. 4), we found that anthropogenic activity factors (nighttime light, nighttime light change, cropland coverage, and cropland change) dominated the changes in FFI during 2000–2020 in the most developed areas, such as the eastern US, Europe, and South China (Supplementary Fig. 5). Moreover, wildfire mainly controlled the $\Delta$FFI of some areas in Canada, Far East Russia, the southeastern Amazon, tropical Africa, and Australia. In addition, for hotspots with decreased FFI (Fig. 3a–c), $\Delta$FFI was most strongly related to wildfire frequency ($P < 0.001$, standardized coefficient = 0.061) in western Canada, while the most important driving factors of $\Delta$FFI in southern Europe and central China were mean cropland coverage (P < 0.001, standardized coefficient = 0.244) and cropland coverage change ($P < 0.001$, standardized coefficient = −0.132), respectively. For hotspots with increased FFI (Fig. 3d–f), wildfire frequency was the strongest driving factor of $\Delta$FFI in the southeastern Amazon ($P < 0.001$, standardized coefficient = 0.299) and central Siberia ($P < 0.001$, standardized coefficient = 0.466), while $\Delta$FFI in the Congo Basin was significantly affected by all factors except nighttime light.

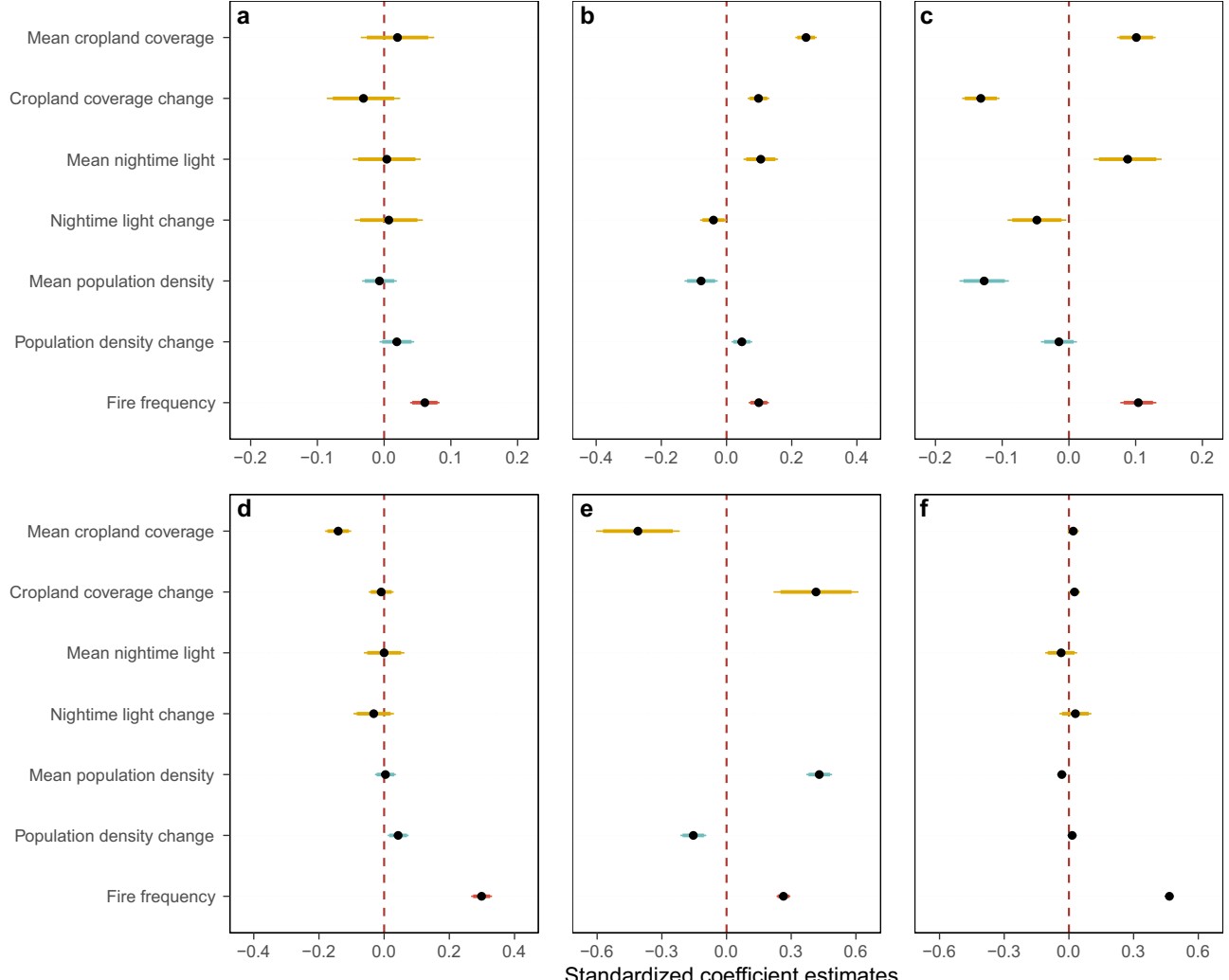

**Fig. 3 | Standardized correlation coefficients of the dynamic forest fragmentation index (ΔFFI) for six hotspots.** Relative effects of anthropogenic activity (the mean and difference values of cropland coverage and nighttime light, yellow color), demographic pressure (the mean and difference values of population density, blue color), and natural disturbance (fire frequency, red color) on dynamics of ΔFFI in (**a–c**) forest fragmentation decreased hotspots (D1–D3, $n = 7661$, 3663, and 5271 forest pixels) and in (**d–f**) forest fragmentation increased hotspots (I1-I3, $n = 4044$, 5104, and 7689 forest pixels). The dots represent standardized coefficient estimates with 95% (thin segments, ±1.960 standard errors) and 90% (thick segments, ±1.645 standard errors) confidence intervals in generalized linear models. Locations of the six hotspots can be checked in Fig. 2.

## A new framework for the assessment of global forest landscape dynamics

As changes in areas and patterns are two key indicators when assessing forest landscape dynamics, we developed a two-dimensional framework based on the changes in FC and FFI between 2000 and 2020 to obtain a comprehensive understanding of forest landscape dynamics. We found that the forest landscapes with a pattern of FC$^{up}$FFI$_{down}$ were distributed worldwide but were concentrated in the western Canada, the northeastern US, northern Eurasia and central China (Fig. 4a). FC$^{up}$FFI$_{down}$ generally accounted for the highest forest area percentage in temperate (50.0%) and boreal (59.2%) regions (Fig. 3b). In contrast, the percentage of forest landscape area that exhibited the FC$_{down}$FFI$^{up}$ pattern in tropical regions (39.8%) was much higher than that in subtropical (27.9%), temperate (14.3%) or boreal (10.6%) regions. Forest landscapes with a pattern of FC$_{down}$FFI$^{up}$ were distributed mostly in the tropics, northern Europe, and central Siberia. Moreover, forest landscapes with patterns of FC$^{up}$FFI$^{up}$ and FC$_{down}$FFI$_{down}$, which accounted for 5.7–7.8% and 24.5–34.2% of the total area, were mainly distributed in northern Europe, the central Amazon, and south tropical Africa respectively.

We also used this framework to assess forest landscapes dynamic patterns for forested countries worldwide. Forest landscapes in most countries generally exhibited the FC$^{up}$FFI$_{down}$ ($n = 40$), FC$_{down}$FFI$^{up}$ ($n = 32$) or FC$_{down}$FFI$_{down}$ ($n = 54$), while those in only five countries predominantly displayed the FC$^{up}$FFI$^{up}$ pattern. At the national scale, the ΔFFI of a country was significantly and negatively correlated with the ΔFC ($R^2 = 0.35$, $P < 0.001$) (Fig. 3c). In addition, among the 10 countries with the largest forest area, the overall forest landscape dynamics pattern was exhibited as either FC$^{up}$FFI$_{down}$ (Russia, China, and India), FC$_{down}$FFI$^{up}$ (Brazil, Australia, the Democratic Republic of the Congo, and Peru), or FC$_{down}$FFI$_{down}$ (Canada, the US, and Indonesia) (Table 1). The FC$^{up}$FFI$_{down}$ pattern was found for more than half of the forest landscapes in Russia (59%), Canada (56%), the US (53%) and China (58%), while a considerable proportion of total forest landscapes in Brazil (42%), the Democratic Republic of Congo (54%) and Peru (43%) had a FC$_{down}$FFI$^{up}$ pattern. In particular, China (ΔFC = 1.21%, ΔFFI = −0.07) had relatively high ΔFC and low ΔFFI values, while Brazil (ΔFC = −3.22%, ΔFFI = 0.014) had relatively low ΔFC and high ΔFFI values. Moreover, for forest landscapes in the world's major forested countries, the FC$_{down}$FFI$_{down}$ pattern occupied high

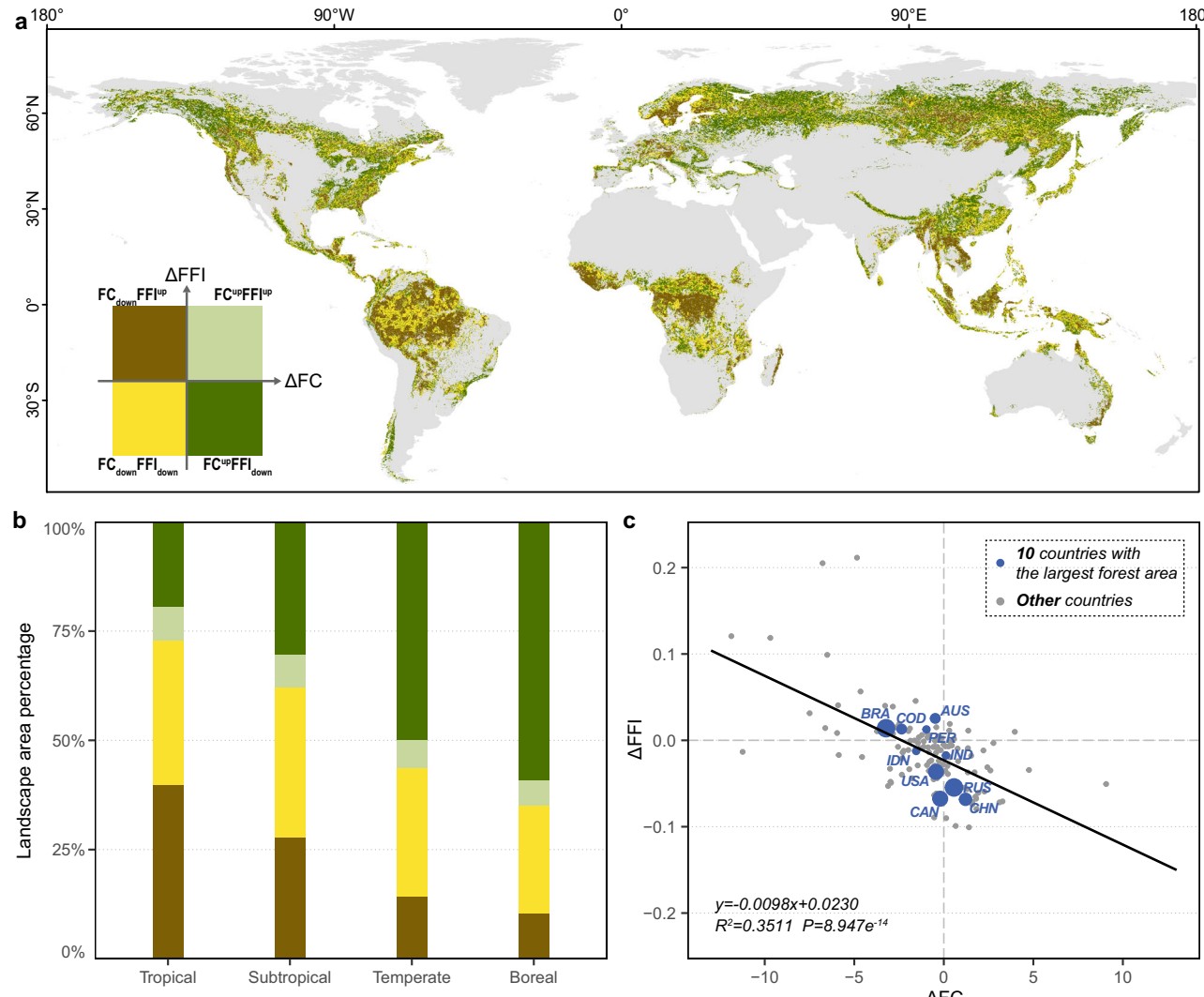

**Fig. 4 | Spatial distributions of different forest landscape dynamic patterns and their area percentages among climatic zones.** Landscape dynamic pattern is defined by the changes in forest fragmentation index (FFI) and forest coverage (FC), and the marks of "up" and "down" after FFI or FC represent an increase and decrease trend during 2000–2020, respectively. **a** Global spatial distribution of four forest landscape dynamic patterns, (**b**) relative area percentages of the four-forest landscape dynamic patterns among climatic zones, and (**c**) the relationship between ΔFFI and ΔFC for forest landscapes at national scale using the Pearson's linear correlation ($n = 131$ countries). The statistical significance in (**c**) was obtained with a two-side Student's T-test.

**Table 1 | Areas, area percentages, and mean values of the changes in forest coverage and fragmentation of four forest landscape dynamic patterns in the ten countries with the largest forest area globally**

| Country | Area and area percentages | | | | | | | | ΔFC | ΔFFI |
|---|---|---|---|---|---|---|---|---|---|---|
| | FC$^{up}$FFI$_{down}$ | | FC$^{up}$FFI$^{up}$ | | FC$_{down}$FFI$_{down}$ | | FC$_{down}$FFI$^{up}$ | | | |
| | $10^5$ km$^2$ | % | $10^5$ km$^2$ | % | $10^5$ km$^2$ | % | $10^5$ km$^2$ | % | Mean | Mean |
| Russian federation | 173.6 | 58.8 | 18.2 | 6.2 | 70.6 | 23.9 | 32.6 | 11.1 | 0.57 | −0.05 |
| Brazil | 7.7 | 16.5 | 3.1 | 6.7 | 16.1 | 34.5 | 19.8 | 42.3 | −3.22 | 0.01 |
| Canada | 52.2 | 56.1 | 4.8 | 5.2 | 26.0 | 28.0 | 10.0 | 10.7 | −0.20 | −0.07 |
| United States of America | 38.3 | 53.0 | 4.1 | 5.6 | 20.1 | 27.8 | 9.9 | 13.6 | −0.43 | −0.04 |
| China | 24.7 | 58.5 | 1.6 | 3.7 | 14.1 | 33.4 | 1.9 | 4.4 | 1.21 | −0.07 |
| Australia | 4.3 | 33.0 | 1.9 | 14.6 | 3.3 | 25.1 | 3.5 | 27.3 | −0.48 | 0.03 |
| Democratic Republic of the Congo | 2.9 | 15.8 | 1.4 | 7.8 | 4.1 | 22.6 | 9.8 | 53.8 | −2.35 | 0.01 |
| Indonesia | 4.3 | 25.6 | 1.5 | 8.8 | 5.8 | 34.7 | 5.2 | 30.9 | −1.54 | −0.01 |
| Peru | 1.2 | 14.7 | 0.7 | 7.9 | 2.8 | 34.1 | 3.6 | 43.3 | −0.97 | 0.01 |
| India | 3.0 | 37.0 | 1.1 | 13.2 | 2.4 | 29.8 | 1.6 | 20.0 | 0.12 | −0.02 |

FC$^{up}$FFI$_{down}$: increased coverage and decreased fragmentation, FC$^{up}$FFI$^{up}$: increased coverage and increased fragmentation, FC$_{down}$FFI$_{down}$: decreased coverage and decreased fragmentation, and FC$_{down}$FFI$^{up}$ decreased coverage and increased fragmentation.

proportions (23% – 35%) of the forested areas, while $FC^{up}FFI^{up}$ pattern only accounted for very low proportions (4–15%).

## Discussion

We developed an integrated FFI and evaluated the static and dynamic patterns of forest fragmentation between 2000 and 2020, and the results identified the world's most fragmented forests and those that experienced the most severe fragmentation. Consistent with previous studies[5,20,21], we found that forest landscapes were relatively intact in the Amazon, the Democratic Republic of the Congo, Borneo, and New Guinea, which host some of the highest biodiversity in the world[22]. However, these areas have also experienced the most severe forest fragmentation over the last two decades according to their higher positive ΔFFI values. For example, tropical regions suffered from more intensive conditions that drove the conversion of intact forests into fragmented forests[7,10–12]. In contrast, although forest landscapes in Europe and South China are highly fragmented, they are recovering as a result of afforestation and effective protection efforts, which has significantly improved forest landscapes. These findings demonstrated that the dynamic FFI, which reflects the nature of fragmentation (changes in forest distribution pattern), may be more appropriate in evaluating forest fragmentation than the static FFI. Our approach separates forest fragmentation from the distribution of forest fragments, which are often conflated in forest fragmentation assessments. The static FFI mainly reveals forest distribution patterns that are the long-term consequence of climate, topography, and historical land cover changes since the onset of the Anthropocene[23–26], and the dynamic FFI represents the processes of forest fragmentation more accurately.

In addition, the two-dimensional assessment framework we developed based on changes in forest fragmentation and forest coverage further reduced the uncertainties in the evaluation of forest landscape dynamic patterns due to the inconsistencies between these two factors. For example, we found that the FFI decreased in some areas in central Canada and the central Amazon between 2000 and 2020, but forest losses were still found in those areas. These findings further emphasize the value of our proposed two-dimensional assessment framework in the evaluation of forest landscape pattern dynamics, and it also provides reasonable approaches for evaluating forest fragmentation and its dynamics at regional-national-global scales. Thus, our approach can be used to link these changes in the spatiotemporal pattern of forest fragmentation to forest management policies[27,28].

We found that 75.1% of global forests experienced a decline in fragmentation during the first 20 years of the 21st century, which suggested that most global forest landscapes were generally improving. However, forest fragmentation exhibited divergent patterns in different regions of the world. On the one hand, we found an extensive decline in forest fragmentation in the world's most densely populated and economically developed regions (the eastern US, Europe, and South China). On the other hand, the remarkable increase in FFI and decrease in FC in tropical areas presumably reflected the fact that forests in these regions are under tremendous pressure from human beings[11,29,30]. If these trends continue, forest fragmentation in the tropics will be further exacerbated, and the ecological functions and values of these forest landscapes will further decline and seriously undermine the role of these forests in international climate agreements and biodiversity conservation.

In our study, various forest landscape dynamic patterns, generated by the two-dimensional assessment framework, reflected different recovery or degradation states of forests worldwide. In the early stage of forest degradation, small or irregular forest patches are cleared first, resulting in an $FC_{down}FFI_{down}$ pattern. As forest loss spreads to large intact patches, more forest edges are created, which causes the forest to exhibit the $FC_{down}FFI^{up}$ pattern and enter a deep degradation stage. In contrast, in the forest recovery (forest protection

and afforestation) scenario, the introduction of additional small patches until they are connected into large intact forest patches causes forests to exhibit $FC^{up}FFI^{up}$ (more forest patches) and $FC^{up}FFI_{down}$ (fewer forest patches) patterns, which represent the initial and deep stages of the forest recovery, respectively.

Our findings indicated that there was a large contrast in the relationship between forests and human beings globally. The widespread deep recovery of forests in some subtropical regions (particularly in South China) and forests with deep degradation in some tropical regions (particularly in the Brazilian Amazon) were both mainly attributed to government forest policies (afforestation and deforestation)[31–35]. For example, the substantial decrease in FFI and increase in FC in central and South China since 2000 have been mainly attributed to the implementation of various ecological protection projects[36–38]. In contrast, the land policy promulgated by the Brazilian government and its associated wildfire disturbances led to sharp forest losses[39] and significantly exacerbated forest fragmentation in the southeastern Amazon. Similarly, large forest losses in southeast Asia and tropical Africa were driven by increasing human pressures according to satellite-based evidence[14,40–42]. However, considering that the early recovery and early degradation stages accounted for considerable percentages of all global forest landscapes, there is a large opportunity for society to improve the forest landscape dynamic patterns by adjusting policies. Particularly in the tropics, timely conservation and restoration efforts will prevent further damage and maintain the important functions of these forests in global biodiversity conservation and climate change mitigation.

Identifying the distribution and composition of forest fragmentation modes enhances our insights for understanding the forest fragmentation processes and drivers. We found that the most typical FFI decrease mode ($ED_{down}PD_{down}MPA^{up}$) and FFI increase mode ($ED^{up}PD^{up}MPA_{down}$) accounted for more than half of the world's relevant forest landscapes, which indicated that edge, isolation, and patch size effects changed synergistically with forest cover change for most global forest landscapes. However, we also found some atypical FFI change modes. For example, in the central Amazon, MPA decreased in some areas where FFI decreased, while PD decreased in some areas where FFI decreased, which indicated that the processes of forest fragmentation were extremely complex. Therefore, efforts in detecting the underlying mechanism of forest fragmentation change should be site-specific, and focus on the relationship between explanatory factors and forest landscape patterns.

By coupling the changes in individual FFI components, we investigated ΔFFI and its associations with anthropogenic and natural factors and identified revealed the possible causes for forest fragmentation dynamics for some hotspots. For hotspots with increased FFI in the southeastern Amazon, large intact forest patches have been converted into multiple small patches under the mixed pressures from commercial harvest, cropland expansion and fire disturbances[39], causing serious forest losses and an increase in fragmentation (Supplementary Fig. 6). In central Siberia, however, forest losses due to fire disturbances, especially in forest edges, directly increased forest fragmentation during 2000–2020. For hotspots with decreased FFI in subtropical regions, especially in central China, the decrease in fragmentation was highly related to the implementation of ecological restoration projects under rapid economic development. For example, afforestation efforts under the "grain to green" project increased forest area and connected discrete forest patches[43]. Also, for hotspots in western Canada and eastern Europe, the decline of FFI was mainly attributed to fire disturbances and changes in cropland area, respectively. These factors increase MPA, reduce ED and PD, and ultimately reduce forest fragmentation by smoothing forest edges and reducing small forest patches.

Although climate change is unlikely to cause changes in forest distribution on a 20-year timescale in most regions of the world, it is

possible that climate may still have certain impacts on forest fragmentation dynamics in some regions. For example, the decrease in FFI in northern Eurasia between 2000 and 2020 may also be attributed to forest expansion caused by climate warming at high latitudes[44], which caused the conversion of small patches into large patches and reduced forest fragmentation. In contrast, more frequent fires, caused by climate change in Canada, Far East Russia, the Brazilian Amazon, tropical Africa and coastal Australia (Supplementary Fig. 5) have resulted in remarkable forest losses and intensified forest fragmentation[45–48].

However, the complexities of forest fragmentation processes and the causes also remind us that forest fragmentation studies should be targeted and localized. The availability of precise forest distribution data and a thorough understanding of forest landscape dynamic drivers, including land policy, climate change, and international trade, are essential conditions for research on the patterns, causes, ecological consequences, and coping strategies of forest fragmentation. Moreover, it should be noted that the use of bi-temporal forest cover data also makes it impossible to fully assess the continuous dynamics, especially in some areas of sub-tropical forestry or humid tropical shifting cultivation. Therefore, more comprehensive analyses should be conducted that consider specific species characteristics, vegetation types, and multi-temporal forest cover data in the detection of the causes of forest fragmentation dynamics. These analyses also form the basis for applying the assessment of patterns and causes of forest fragmentation to biodiversity conservation and carbon cycle feedback mechanisms.

By coupling landscape changes and multiple features of fragmentation, our approach overcomes the problem of considering only the static state of fragmentation instead of dynamic change, and thus it improves our understanding of the patterns of global forest fragmentation. It also more effectively reflects the reality of forest landscape changes and is valuable for the timely formulation and adjustment of relevant policies.

In addition, negative states of forests exist for most of the 10 countries with the largest forest areas, which demonstrates remarkable agricultural land expansion, timber harvest and forest fire disturbance in recent years, as well as an overall increase in other disturbances. These changes result in the further loss of forest area, the intensification of fragmentation, and the degradation of ecosystem functions[28,49,50]. However, the significant negative relationship between ΔFFI and ΔFC at the national scale suggested that efforts that aimed to increase forest area were still effective in mitigating fragmentation. Targeted afforestation and protection measures are important approaches to prevent further deterioration of fragmentation globally. Our findings highlight that an understanding of deforestation and fragmentation dynamics needs to be incorporated into the policy-making process in these countries to minimize irreversible damage to vital forest ecosystems and to redirect the course of development toward sustainability.

## Methods

### Datasets
**Global forest cover maps.** High-resolution (30 m) global forest cover data for 2000 and 2020 were obtained from the global land cover and land use (GLCLU) change dataset[51]. The GLULC forest cover data defines a pixel with tree height ≥5 m at the Landsat pixel scale as a forest pixel, which agrees with the definition of forest by the Food and Agriculture Organization of the United Nations (FAO). The 30 m forest cover data were processed into binary forest maps for 2000 and 2020 and were used to calculate three fragmentation-related landscape metrics based on the global 5000 m grid (see calculation details below). The 5000 m grid was also used to calculate the forest coverage (FC, the percentage of forest area to total area in a particular space) in each grid cell and to generate 5000 m resolution FC maps for 2000 and 2020.

**Climatic zones.** The climate zones data were from the world climate regions (WCR) map produced in 2020 (Supplementary Fig. 7a)[52], with a spatial resolution of 250 m. We considered global forests to occur in one of four zones: tropical, subtropical, temperate or boreal, which represent the major climatic zones of relevance to forest distribution and spatial patterns of forest fragmentation.

**Digital elevation model (DEM).** Raster-based global altitude data were obtained from the Global Land One-kilometer Base Elevation (GLOBE) dataset (Supplementary Fig. 7b)[53]. GLOBE is a global digital elevation model (DEM) with a latitude-longitude grid spacing of 30 arc-seconds. The GLOBE DEM dataset was aggregated to a spatial resolution of 5000 m to match the FFI and FC data, and the continuous DEM data were divided into 12 altitudinal categories (<0, 0–100 m, 100–200 m, 200–300 m, 300–400 m, 400–500 m, 500–600 m, 600–700 m, 700–800 m, 800–900 m, 900–1000 m, and >1000 m) to analyze the relationship between forest fragmentation and altitude.

### Estimates of global forest fragmentation
**Global grid extent.** We developed a series of grids, 5000 m × 5000 m in size, for 2000 and 2020 to cover the global forest area. For each grid cell, the fragmentation-related landscape pattern metrics were calculated based on the forest/non-forest binary maps. In total 3,413,077 and 3,422,375 grid cells were considered for 2000 and 2020, respectively, which mainly covered all forest landscapes worldwide. The total area of forest landscapes was larger than the actual global forest area, because each forest landscape contains a certain proportion of non-forest areas. The overlaid parts of the grids were ultimately used in this study to analyze the changes in forest fragmentation.

**Forest landscape metrics related to fragmentation.** Edge effect, isolation effect, and patch size effect were the most important features of forest fragmentation[5], and they can be quantified by three landscape pattern metrics, including edge density (ED), patch density (PD) and mean patch area (MPA), respectively. The three landscape pattern metrics were used to assemble a synthetic forest fragmentation index in our study. These metrics were calculated as follows:

$$ED = \frac{\sum_{k=1}^{m} e_{ik}}{A} * 10,000 \tag{1}$$

$$PD = \frac{n_i}{A} * 10000 * 100 \tag{2}$$

$$MPA = mean\left(AREA\left[patch_{ij}\right]\right) \tag{3}$$

where $e_{ik}$ is the total edge length in meters, $n_i$ is the number of patches, $A$ is the total landscape (a grid cell was regarded as a landscape) area in square meters, and AREA [$patch_{ij}$] is the area of each patch in hectares.

These three landscape pattern metrics were calculated at the class level (class 0: non-forest, class 1: forest) using the "landscapemetrics" package[54] in R software based on the two binary maps described above, and the values were converted into raster layers for subsequent analysis.

**Static and dynamic forest fragmentation indexes.** To obtain a comprehensive understanding of the multiple dimensions of fragmentation, we constructed a synthesized forest fragmentation index (FFI) using the normalized single fragmentation metrics in ArcGIS software. During the normalization of ED, PD, and MPA, both the directions of the three metrics in reflecting forest fragmentation and the comparability of the FFI in different years were considered (Supplementary Note 1 in Supplementary Information). We used the

difference in the FFI between 2020 and 2000 to construct the dynamic forest fragmentation index (ΔFFI) using the follow equation:

$$\triangle FFI = FFI_{2020} - FFI_{2000} \qquad (4)$$

where the range of ΔFFI is −1-1. The negative and positive values of ΔFFI indicated decreased and increased fragmentation, respectively.

**Spatial patterns of forest fragmentation.** To evaluate the spatial patterns of the static FFI and ΔFFI values, we compared the $FFI_{2000}$, $FFI_{2020}$, and ΔFFI among the different climatic zones and altitudinal categories defined previously. One-way ANOVA was used to test for significant differences in $FFI_{2000}$, $FFI_{2020}$, and ΔFFI values, and the LSD test was used for pairwise comparisons (Fig. 1d). Moreover, a generalized additive model (GAM) was used to detect the relationships between ΔFFI and $FFI_{2000}$ for the different climatic zones (Supplementary Fig. 2). Linear correlations were used to explore the relationships between altitude and $FFI_{2000}$, $FFI_{2020}$, and ΔFFI, and the correlation coefficient and the slope of the regression line were both used to assess the strength of these relationships.

**Spatial patterns of the modes of fragmentation processes**
Since the ΔFFI was calculated by combining ED, PD and MPA, the directions of change in these metrics can reflect the different processes forest fragmentation can proceed. Therefore, we identified eight modes of forest fragmentation processes ($ED^{up}PD^{up}MPA_{down}$, $ED_{down}PD^{up}MPA_{down}$, $ED^{up}PD_{down}MPA_{down}$, $ED_{down}PD_{down}MPA_{down}$, $ED^{up}PD^{up}MPA^{up}$, $ED_{down}PD^{up}MPA^{up}$, $ED^{up}PD_{down}MPA^{up}$, and $ED_{down}PD_{down}MPA^{up}$) by considering all possible combinations of an increase or decrease in the ED, PD and MPA between 2000 and 2020 (Fig. 2). Moreover, to characterize the forest fragmentation processes that occurred in areas where fragmentation decreased or increased, we evaluated the global spatial distributions and composition proportions of the different fragmentation process modes where the ΔFFI values were negative or positive, respectively.

To better analyze the processes driving changes in forest fragmentation, we selected three hotspots (western Canada, southern Europe, and central China) where fragmentation was remarkably decreased and three hotspots (the southeastern Amazon, the Congo Basin, and central Siberia) where fragmentation was obviously increased. For each hotspot, we analyzed the composition proportions of the eight modes of fragmentation processes (Fig. 2). We then used a T-test to compare the values of $ED_{nor}$, $PD_{nor}$ and $MPA_{nor}$ between 2000 and 2020, and the percentages of increase or decrease for each of these values were used to reflect their contributions to the changes in the FFI.

**Drivers of forest fragmentation processes for the globe and hotspots**
**Explanatory variables.** Anthropogenic activities, demographic pressure, and natural disturbances are considered as the main drivers of global forest loss[39]. Considering the relationship between human and forest cover, anthropogenic activities were further divided into agricultural activity and socio-economic intensity. Therefore, factors regarding agricultural activity (mean cropland coverage and cropland coverage change), socio-economic intensity (mean nighttime light and nighttime light change), demographic pressure (mean population density and population density change), and natural disturbance (fire frequency) were adopted in our study to explain the dynamics of FFI for six hotspots and the globe. Agricultural activity and natural disturbance are direct influencing factors for the changes in forest distribution and forest fragmentation, while socio-economic intensity and demographic pressure affect forest cover and forest fragmentation by indirect pathway.

For variables of agricultural activity, the mean cropland coverage and cropland coverage change were regarded as two important indicators that represent the magnitude and variation of cropland area during 2000–2020. The 30 m resolution cropland extent maps from the Global Land Cover and Land Use Change dataset[55] for 2003 and 2019 were adopted in our study and were processed into cropland coverage data by calculating the ratio of cropland pixel to total pixel in a 5000 m size grid. The mean and the difference values of the cropland coverage between 2003 and 2019, respectively, were directly used as agricultural activity factors. In addition, the 500 m resolution nighttime light data, derived from the global NPP-VIIRS-like nighttime light dataset[56], for 2000 and 2020 were directly used to represent the socio-economic intensity factors. Similarly, the mean nighttime light and nighttime light change were two important indicators that represent the magnitude and variation of socio-economic intensity during 2000–2020, respectively, and were calculated by the mean and difference values of the NPP-VIIRS-like nighttime light between 2000 and 2020. Furthermore, the mean population density and the population density change during 2000–2020 reflect the magnitude and variation of demographic pressure, respectively, and were used as demographic pressure factors in this study. The WorldPop global gridded population count datasets[57] for 2000 and 2020 with a spatial resolution of 1000 m were firstly aggregated into 5000 m resolution population density data by calculating the mean value for each 5000 m size grid of the two periods. The two demographic pressure variables were obtained by the mean and the difference values of the 5000 m resolution gridded population density layers for 2000 and 2020. Finally, for the natural disturbance variable, fire frequency during 2000–2020 was selected as an important indicator that represented the total fire disturbance. The MODIS monthly global burned area dataset (MCD64A1 version 6) with a spatial resolution of 500 m were used in this study, and the fire frequency was obtained by counting the ratio of burned pixel during 2000–2020 for 5000 m size grids during 2000–2020. The resolution of raster layers of all independent variables was aggregated into 5000 m to match the ΔFFI.

**Correlation analysis.** The general linear models were used to detect the relationship between ΔFFI and the seven explanatory factors, and all dependent and independent variables were standardized into the range of 0-1 during statistical analysis. The impact of each explanatory factor on the ΔFFI was quantified by the standardized coefficient estimates and P values from the standardized multiple linear models, and the corresponding confidence intervals were also incorporated to help analyze the drivers of ΔFFI. We performed the general linear models for each hotspot (Fig. 3) and the globe (Supplementary Fig. 4) based on dependent and independent variables in their respective scopes. In addition, we identified the major driver of ΔFFI, represented by the factor with the highest absolute value of coefficient estimates, for a series of 50 km × 50 km grids and finally generated the ΔFFI major driver map (Supplementary Fig. 5) with a spatial resolution of 50 km at the global scale.

**Two-dimensional framework for the assessment of forest landscape dynamics**
We constructed a two-dimensional assessment framework of global forest landscape dynamics in which all forest landscapes were categorized into four types ($FC^{up}FFI_{down}$, $FC^{up}FFI^{up}$, $FC_{down}FFI_{down}$, and $FC_{down}FFI^{up}$) based on increases (positive values) or decreases (negative values) in FC and FFI from 2000 to 2020. These four types of forest landscape dynamic patterns represent forest landscapes in the deep recovery stage, early recovery stage, early degradation stage and deep degradation stage, respectively. We mapped the spatial distributions of these different forest landscape dynamic patterns (Fig. 3a) by overlaying the ΔFFI layer and the ΔFC layer at 5000 m resolution.

We also evaluated the variation in the relative percentages of the four patterns across climatic zones (Fig. 3b) and the altitudinal gradient (Supplementary Fig. 8). Furthermore, we compared the mean values of ΔFFI and ΔFC among all countries worldwide (Supplementary Data 1) and used Pearson's linear correlations to fit the relationship (Fig. 3c). Specifically, to evaluate how changes in forest cover and fragmentation between 2000 and 2020 varied among the ten countries with the largest forest area worldwide, we calculated the area percentages representing each pattern and the mean values of ΔFC and ΔFFI (Table 1). The administrative boundaries of each country were determined by the FAO (https://data.apps.fao.org/map/catalog/static/search?format=shapefile).

### Reporting summary

Further information on research design is available in the Nature Portfolio Reporting Summary linked to this article.

## Data availability

All data used in the analysis are publicly accessible. The global land cover and land use change dataset is available at https://glad.umd.edu/dataset/GLCLUC2020 (including forest cover data and cropland coverage data); The global climate zones dataset is available at https://storymaps.arcgis.com/stories/61a5d4e9494f46c2b520a984b2398f3b; The global altitude dataset is available at https://ngdc.noaa.gov/mgg/topo/gltiles.html; The global NPP-VIIRS-like nighttime light dataset is available at https://dataverse.harvard.edu/dataset.xhtml?persistentId=doi:10.7910/DVN/YGIVCD; The worldPop global gridded population count dataset is available at https://hub.worldpop.org/project/categories?id=3; The MODIS monthly global burned area dataset is available at https://lpdaac.usgs.gov/products/mcd64a1v006/. The global administrative boundary dataset is available at https://data.apps.fao.org/map/catalog/static/search?format=shapefile. The Forest Fragmentation Index (FFI) data generated in this study have been deposited in the Figshare repository at https://figshare.com/s/21dbf1f50250aeb7f5a0.

## Code availability

The code used to calculate the landscape pattern index in this study can be found in the Figshare repository at https://figshare.com/s/21dbf1f50250aeb7f5a0.

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

## Acknowledgements

This research was supported by the National Key Research and Development Program of China (2022YFF0802400, J.M.) and the Natural Science Foundation of China (32271659, J.M. and U2106209, J.M.).

## Author contributions

J.M. and J.J.L. conceived the idea and designed the methodology; J.M., J.W.L., and W.B.W. conducted the data analysis; and J.M. wrote the manuscript with contributions from J.W.L., and J.J.L. All authors contributed critically to the interpretation of the results and gave final approval for publication.

## Competing interests

The authors declare no competing interests.
