## [Peer review file · Nature Communications]

REVIEWER COMMENTS

Reviewer #1 (Remarks to the Author):

The authors present a global analysis of forest fragmentation dynamics across all terrestrial forest biomes, using high-resolution satellite estimation of forest cover changes between 2000 and 2020. These analyses integrating the temporal dimension of global forest fragmentation seem pretty new, timely very needed to understand the ongoing changes in spatial patterns of forest cover. Most “global” past studies have focused on the fragmentation dynamics of tropical forests, while here, the authors present an accurate global analysis that englobes boreal, temperate, tropical, and subtropical forests. Even though I’m not a specialist in forest fragmentation, I value the approach aiming at simultaneously monitoring 1) changes in a general fragmentation index and 2) different path/dynamics trajectories in more precise metrics (i.e., edge density, patch density, mean patch area). I also find it very insightful to link changes in fragmentation with changes in total forest cover. However, I have a major concern, which I will develop below, about the authors' choice of forest cover data. This concern must be seriously tackled for considering a publication in a generalist high-impact journal such as Nature Communications.

My major concern is the authors’ choice of raw forest cover data. The authors use two satellite-based data sets to assess changes in forest fragmentation between 2000 and 2010. The first one (i.e., 2000) is the well-known Hanssen et al. (2013) forest cover product, which estimates the tree cover fraction (%) within 30 m resolution pixels across the globe. This dataset is based on Landsat images and covers the 2000-2021 period with a one-year temporal resolution (see global forest change dataset). So, this dataset was available for the two periods that the authors analyzed (2000 and 2020). But the authors decided to switch to a different dataset for the 2020 period, which is the ESA Landcover data, which classifies the global land cover in categories (including forests) in 10 m resolution pixels with Sentinel images. Because the two datasets have different spatial resolutions (30m and 10 m) and ecological resolutions (continuous and categorical), the data must pass through two different major transformations to obtain two comparable binary forest cover maps (i.e., forest vs non-forest) at 30 m resolution. The Hanssen dataset is transformed to the categorical using threshold (pixels with forest cover >30% are classified as forest), and the ESA Worldcover categorical data are aggregated to 30 m resolution (we lack crucial information about this process in the methods section). There is no justification that the two transformed datasets are comparable, which is a major point considering that the analyses focus on fine-scale spatial patterns in forest cover. Finally, I do not understand why the authors did not choose to keep the Hanssen data set for the two time periods, and the authors must justify this choice.

I also have more minor concerns/comments/suggestions:

Building on the major concern, I also want to note that there would be significant advantages to using the Hanssen (2013) dataset for the two periods. Using this dataset in 2000 and 2020 would help to consider subtle temporal-scale forest cover changes within grid cells. With the yearly temporal resolution, each 30m pixels can be classified as having recording loss, gains or both loss & gains between the two time periods, which may help explain fragmentation dynamics trends.

Maybe my calculus is wrong, but I think there is a problem with the global forest cover extent calculations. The authors say (lines 384-389) that the $\approx 280,000 \times 25\text{km}^2$ grid cells contain almost all forest landscapes worldwide. However, a simple multiplication shows that it results in ≈ 7 million km^2 (700 million hectares) of forest cover worldwide, while the current FAO estimation for 2020 is about 4 billion hectares (FAO report).

I understand that abbreviations can be practical, but the authors could reduce the number of acronyms within the text (or at least give the abbreviations full names within figure captions). More broadly, adding details on “how to interpret” the results in the figure captions would greatly help

readers follow the text.

Curtis et al. (2018, Science) have classified the main drivers of forest loss worldwide. The authors should cite and discuss their results with this previous work to link their results (trends in fragmentation dynamics) to drivers in different regions. They even could integrate the Curtis et al. (2018) results in their analyses.

I appreciate the figures, but I feel they could be better quality (e.g., vectorial).

Reviewer #2 (Remarks to the Author):

I find the study to be a valuable approach to characterizing global forest change in the spatial domain. Examining forests fragmentation dynamically is the only way to go – we no longer need static maps. The approach is clear, although the shorthand (metric abbreviations) is a bit too dense to quickly uptake and interpret.

I find two major limitations of the study. First, the post-characterization comparison of two very different maps will result in incorrect change estimation. In remote sensing, the overlaying of two thematic maps contains true change and, of course, false change due to errors in either of the input maps (in most cases the two maps are made from similar data inputs, using the same method and the same definitions of themes of interest). Things get more problematic when comparing two very different thematic maps as the basis of comparison. The use of a 30m Landsat-based 2000 percent tree cover map compared to a discrete 10m Sentinel 2-based forest/non-forest map will have regional differences due to the varying methods, spatial scales, and nominal forest definitions used. When mapping forests, especially in areas where the formations are on the edge of the nominal definition, you will get false change. I believe boreal change in Central Russia could be one such example – a region where much of the landscape is on the edge of the physiognomic/structural definition of forest may yield different results in different mapping exercises. In any case, the definition of forest should be included – what is it in terms of height cover, minimum mapping unit? Are you able to truly harmonize the different maps, and if not, what is the impact or qualifications on the results? The paper presents all results as valid and unaffected by such impacts/considerations. The text describes findings with considerable qualified language regarding changes that “may be due to...” or “suggests that...” This is highly problematic as not all change may in fact be real. Some quantification of this issue is required, which comes to the second critique.

Attributes/drivers of forest fragmentation should be quantified, mainly through a sample-based assessment of the different categories at regional scales. The discussion of regional variations is too cursory and unsupported by data, when it could in fact be supported. If you were to examine a sample of each of your categories, you could verify which are definitive and which are not. It is always incumbent on data producers to quantify uncertainties – where in the map are there results that are not supported by reference data, and likely a result of input data limitations? Along these lines, what might you omit or misinterpret using a bi-temporal comparison? Many regions have continuous dynamics, for example sub-tropical forestry, or humid tropical shifting cultivation. How to account for such regions in interpreting gain/loss dynamics? Is mitigation the correct term if forest edge is lessened under a forestry land use regime? I am thinking of the Southeast USA, which I do not think has experienced any positive outcome regarding forest extent/arrangement in recent years, but is instead the site of highly intensive forestry land use. In general, the study, while a neat examination of dynamics, as it should be, does not look ‘under the hood’ enough to qualify or confirm findings.

A couple of minor queries. Why 5000km²? Did you assess various scales of aggregation or was this simply the most computationally doable resolution? What impact does the choice of output grid cell size have on results? Also, please explain more clearly to readers how fragmentation can change

without forest loss or gain. I am thinking of the line where FFI “increased in central Europe between 2000 and 2020, 232 in which no remarkable forest cover change (loss or gain) was detected.” You should show some examples of the various dynamics at full resolution, meaning native map input resolutions, to facilitate understanding of your coarser scale metrics, something along the lines of a tutorial. You could really help users get from the inputs to the metrics through some detailed examples.

Reviewer #3 (Remarks to the Author):

General comments

This study analyzed the changes in the forest fragmentation index in a period to obtain the dynamic fragmentation index and its global pattern. It quantified global patterns of forest fragmentation from a perspective more consistent with the concept of forest fragmentation that cannot be reflected by the static fragmentation index. By analyzing the combination of changes in forest fragmentation and forest coverage, this study developed a framework to quantify the spatial distribution of global forest landscape dynamic patterns (degradation or restoration), discussed the important ecological consequences of forest fragmentation, and provide possible mitigation approaches. This study tries to examine forest fragmentation from a dynamic perspective (fragmentation itself has the nature of change) and prove through quantitative analysis that dynamic FFI can better reflect forest landscape changes in many hot spots around the world than static FFI. I believe this study has important merit in the field of forest landscape dynamic evaluation, and the related forest fragmentation and forest landscape dynamic maps are potentially valuable to stakeholders in both basic research and policy-oriented roles. The article is well-written and the logic is clear.

However, I have several major points I think should be considered before its consideration for publication.

1) The quality, consistency in definition and classification approaches, and the accuracies of two baseline forest maps are key for the reliability of the results. How can the authors confirm that the forest information from the Hansen forest map and ESA WorldCover map can be compared directly? Why are the results not the artifacts of the biased forest maps-based analyses?

2) Three landscape pattern indexes (edge density, patch density, and mean patch area) were considered for the construction of the integrated forest fragmentation index (FFI). According to the global maps of the three indexes from the supplementary information, we found these three individual landscape metrics have similar spatial patterns. So, it is necessary to use three indexes to construct FFI? If so, a more detailed reason for the selection of the three metrics should be provided. In addition, why did the authors use equal weights (if I understand correctly) of the three metrics in calculating FFI?

3) The attribution of the global forest fragmentation changes were kind of weak, and lack of quantitative analyses. I would suggest the authors build a model to consider a few natural and socioeconomic drivers (e.g., climate change, population, cropland expansion, plantations, etc.) in different countries. Also, percolation theory has been considered in the global patterns of tropical forest fragmentation (Taubert et al., Nature 2018), Has this study considered those influence factors in the previous study?

4) In the analysis of forest fragmentation process modes obtained based on the changes of three individual fragmentation-related landscape metrics, the possible meanings represented by atypical modes (modes except A for forest fragmentation exacerbation area and modes except for H for forest fragmentation mitigation area) are not discussed enough. Furthermore, the reasons for the selection of the six hotspots during the analysis of the fragmentation process pattern need to be strengthened. In addition, the authors can think about moving Fig S3 into the main text given the truth that it can demonstrate changes in various metrics in different fragmentation processes.

Specific comments

Line 45: What do you mean by the “first several decades”? The community would agree that the most recent few decades are the best stage for understanding forest fragmentation due to the availability of remote sensing-based forest maps.

Line 80: There might be an error here. Change 0.2 to 0.8?

Figure 3a: in the legend, “FC” can be moved ahead of “FFI”

Line 387: why are the grids in the two years different?

Response to Reviewer's Comments

Dear reviewers,

Many thanks for your time in reviewing our manuscript and providing insightful comments and suggestions. We have carefully read your comments and addressed all these issues in our revised manuscript. Below are our detailed responses and revisions.

(Black: reviewer comments; Blue: our responses; *Purple Italic: our revisions*).

Reviewer #1 (Remarks to the Author):

The authors present a global analysis of forest fragmentation dynamics across all terrestrial forest biomes, using high-resolution satellite estimation of forest cover changes between 2000 and 2020. These analyses integrating the temporal dimension of global forest fragmentation seem pretty new, timely very needed to understand the ongoing changes in spatial patterns of forest cover. Most “global” past studies have focused on the fragmentation dynamics of tropical forests, while here, the authors present an accurate global analysis that englobes boreal, temperate, tropical, and subtropical forests. Even though I'm not a specialist in forest fragmentation, I value the approach aiming at simultaneously monitoring 1) changes in a general fragmentation index and 2) different path/dynamics trajectories in more precise metrics (i.e., edge density, patch density, mean patch area). I also find it very insightful to link changes in fragmentation with changes in total forest cover.

Response: Thank you very much for your positive and valuable comments and suggestions. We have revised this manuscript carefully following your suggestions and comments. We hope the revision can meet with approval.

However, I have a major concern, which I will develop below, about the authors' choice of forest cover data. This concern must be seriously tackled for considering a publication in a generalist high-impact journal such as Nature Communications.

Response: Thank you for your comment. Both you and the other two reviewers mentioned the issue of using inconsistent sources of forest cover data. We have made some explanations on the use of forest cover data in the following responses, and we have changed the forest cover data source in the revision. Forest extent maps for 2000 and 2020 in Global Land Cover and Land Use Change (GLCLU) dataset from a new study of Hansen's team were adopted, and we hope the consistent source of forest cover data in 2000 and 2020 can enhance our results.

My major concern is the authors' choice of raw forest cover data. The authors use two satellite-based data sets to assess changes in forest fragmentation between 2000 and 2010. The first one (i.e., 2000) is the well-known Hansen et al. (2013) forest cover product, which estimates the tree cover fraction (%) within 30 m resolution pixels across the globe. This dataset is based on Landsat images and covers the 2000-2021 period with a one-year temporal resolution (see global forest change dataset). So, this dataset was available for the two periods that the authors analyzed (2000 and 2020). But the authors decided to switch to a different dataset for the 2020 period, which is the ESA Landcover data, which classifies the global land cover in categories (including forests) in 10 m resolution pixels with Sentinel images. Because the two datasets have different spatial resolutions (30m and 10 m) and ecological resolutions (continuous and categorical), the data must pass through two different major transformations to obtain two comparable binary forest cover maps (i.e., forest

vs non-forest) at 30 m resolution. The Hanssen dataset is transformed to the categorical using threshold (pixels with forest cover >30% are classified as forest), and the ESA Worldcover categorical data are aggregated to 30 m resolution (we lack crucial information about this process in the methods section). There is no justification that the two transformed datasets are comparable, which is a major point considering that the analyses focus on fine-scale spatial patterns in forest cover. Finally, I do not understand why the authors did not choose to keep the Hanssen data set for the two time periods, and the authors must justify this choice.

Response: We appreciate your valuable comments. We have made some explanation about the reasons of using forest cover maps in 2000 and 2020 from different data source. In the revision, we have changed the forest cover data and used forest extent maps from the global land cover and land use (GLCLU) change dataset instead.

At the very beginning of this study (June 2021), we planned to use multiple years of forest cover data that derived from Hansen's tree cover map. However, we can only find tree cover map for 2000, forest losses layers for 2001-2021, and forest gains layers for 2001-2012 in the datasets (include updated datasets) related to Hansen's Science paper (Global Land Analysis & Discovery, <https://glad.umd.edu/dataset>). Based on these data, we cannot obtain the tree cover map for 2020. We also reached out to Dr. Matthew Hansen to inquire about tree cover data for 2020 or adjacent years using the same method as the tree cover map of 2000, but unfortunately, we didn't receive a positive response. Actually, we also attempted to obtain consistent forest cover maps for 2000 and 2020 from other data sources, such as the Global Forest Cover Change (GFCC) dataset. However, the publicly available GFCC tree cover dataset only covers the period between 2000 and 2015. We noticed the GFCC tree cover map in 2020 may be available from a commercial website (www.terraPulse.com). We sent messages on the website about how to get the GFCC tree cover data in 2020, but we got no response. Considering that the global forest cover and forest landscape pattern may have large changes during 2015-2020, especially in tropical areas (e.g. Brazilian Amazon), we decided to use forest cover data in 2020 derived from ESA land cover map instead, and to compare it with forest cover data in 2000 derived from Hansen's tree cover map in our forest fragmentation change analysis.

However, after carefully considering the comments from three reviewers, we agree that using forest cover maps from different data sources can be problematic. Fortunately, we discovered a study by Hansen's team on global land cover and land use (GLCLU) change during 2000-2020 (Potapov et al., 2022), which released forest extent maps for 2000 and 2020. The definition of forest set as "a pixel with tree height ≥ 5 m at the Landsat pixel scale" in GLULC forest maps, which is consistent with the definition by the Food and Agriculture Organization of the United Nations (FAO). Therefore, we used the two forest cover maps for 2000 and 2020 from the GLCLU dataset to replace the previous ones.

Based on the new forest cover data, we calculated the three components of forest fragmentation index (FFI), including edge density (ED), patch density (PD), and mean patch area (MPA) at 5km scale, regenerated the static and dynamic FFI distribution maps, and revised the results about the spatial patterns of forest fragmentation and its change. We also revised the contents about the modes of forest fragmentation processes and the assessment framework of forest landscape dynamics according to the new forest cover data.

We acknowledge that there are some small differences in the magnitudes and distributions of the forest fragmentation index (FFI) after using the updated datasets. However, the main results and

conclusions of our study remain unchanged. We have included all changes and revisions in the revised manuscript.

[REF]

Potapov, P. et al. The Global 2000-2020 Land Cover and Land Use Change Dataset Derived From the Landsat Archive: First Results. *Front. Remote Sens.* 3, 856903 (2022).

I also have more minor concerns/comments/suggestions:

Response: Thank you for your insightful comments and suggestions. We have addressed the identified issues and revised the manuscript accordingly.

Building on the major concern, I also want to note that there would be significant advantages to using the Hansen (2013) dataset for the two periods. Using this dataset in 2000 and 2020 would help to consider subtle temporal-scale forest cover changes within grid cells. With the yearly temporal resolution, each 30m pixels can be classified as having recording loss, gains or both loss & gains between the two time periods, which may help explain fragmentation dynamics trends.

Response: Thank you for your comment. We have provided an explanation for our decision not to use Hansen's tree cover map in our earlier response. We also acknowledge the usefulness of Hansen's dataset for analyzing forest cover, losses, and gains in interpreting changes in forest landscape patterns. However, due to the lack of tree cover data for 2020 and forest gains data for certain years between 2013 and 2020, we were unable to obtain forest cover data for 2020 directly or indirectly. Therefore, we had to choose an alternative forest cover product for our analysis of forest fragmentation change from 2000 to 2020. We have provided detailed information about the revisions made to the results based on the new forest cover data in our earlier response.

In addition, we agree that it is important to investigate the causes of changes in forest fragmentation to enhance our study. In response to your suggestion, we have conducted an analysis of changes in the FFI components (edge density, patch density, and mean patch area) during 2000-2020 in six hotspots selected for regions with significant FFI changes (three in areas with increased FFI and three in areas with decreased FFI, respectively) (Fig. 2 in revised version). We also explored the relationship between Δ FFI and explanatory variables, including agricultural activity (mean cropland coverage and cropland coverage change), natural disturbance (fire frequency), socio-economic intensity (mean nighttime light and nighttime light change), and demographic pressure (mean population density and population density change). We used general linear models to identify the significance (p-value) and importance (standardized coefficient) of each factor in this driving force analysis, which was conducted in six hotspots (Fig. 3 in revised version) and all 50 km size grids globally (Fig. S4 and S5 in revised version).

Fig. 2 | Spatial distributions and composition proportions of eight forest fragmentation process modes for forest fragmentation decreased areas (the upper row) and forest fragmentation increased areas (the below row). ED, PD, and MPA mean the individual components of the synthetic forest fragmentation index (FFI). The marks of “up” and “down” after each FFI component represent an increase and decrease trend during 2000-2020, respectively. Three hotspots in the most obvious FFI decreased and increased areas were selected respectively to evaluate the changes in each component of FFI. Hotspots D1-D3 were in western Canada, southern Europe, and central China, and hotspots I4-I6 were in the southeastern Amazon, the Congo Basin, and central Siberia, respectively.

Fig. 3 | Standardized correlation coefficients of the dynamic forest fragmentation index (Δ FFI) for six hotspots. Relative effects of anthropogenic activity (the mean and difference values of cropland coverage and nighttime light, yellow color), demographic pressure (the mean and difference values of population density, blue color), and natural disturbance (fire frequency, red color) on dynamics of Δ FFI in (a)-(c) forest fragmentation decreased hotspots (D1-D3, n=7661, 3663, and 5271) and in (d)-(f) forest fragmentation increased hotspots (I1-I3, n=4044, 5104, and 7689). Dots represent standardized coefficient estimates with 95% (thin segments, ± 1.960 standard errors) and 90% (thick segments, ± 1.645 standard errors) confidence intervals in generalized linear models. Locations of the six hotspots can be checked in Fig. 2.

Fig. S4 | Standardized correlation coefficients of the dynamic forest fragmentation index (Δ FFI) for the globe. Relative effects of anthropogenic activity (the mean and difference values of cropland coverage and nighttime light; yellow color), demographic pressure (the mean and difference values of population density; blue color), and natural disturbance (fire frequency; red color) on dynamics of Δ FFI of all forested countries (n=131). Dots represented standardized coefficient estimates with 95% (thin segments, ± 1.960 standard errors) and 90% (thick segments, ± 1.645 standard errors) confidence intervals in generalized linear models.

Fig. S5 | Primary drivers of the dynamic forest fragmentation index (Δ FFI) for the period 2000 to 2020. The primary driver factor for each 50 km grid was represented by the factor with the highest absolute value of standardized coefficient estimates.

Results regarding the causes of forest fragmentation change during the period of 2000-2020 can be found in the "*Modes and causes of forest fragmentation processes*" section of the revised version.

Line 131 to 169

“We also detected the changes in individual fragmentation-related metrics for hotspots in areas with decreased and increased FFI, respectively. Overall, hotspots where FFI decreased had declines in ED and PD, and increases in MPA. Specifically, MPA increased significantly by 73% in western Canada, by 38% in southern Europe, and by 50% in central China from 2000 to 2020 (Fig. 2). Conversely, increased ED and PD played a more important role in hotspots where FFI increased. ED increased dramatically by 41% in the southeastern Amazon, by 81% in the Congo Basin, and by 90% in central Siberia, while the increments of PD in these hotspots were 32%, 186%, and 78%, respectively. However, MPA decreased slightly (-8% to -31%) in these hotspots where forest fragmentation increased.

Using generalized linear models, we further explored the relationships between Δ FFI and explanatory factors (see Methods) for the globe and the six hotspots. Although Δ FFI was not significantly correlated with any explanatory variables at the global scale (Supplementary Fig. 4), we found that anthropogenic activity factors (nighttime light, nighttime light change, cropland coverage, and cropland change) dominated the changes in FFI during 2000-2020 in the most developed areas, such as the eastern US, Europe, and South China (Supplementary Fig. 5). Moreover, wildfire mainly controlled the Δ FFI of some areas in Canada, Far East Russia, the southeastern Amazon, tropical Africa, and Australia. In addition, for hotspots with decreased FFI (Fig. 3a-c), Δ FFI was most strongly related to wildfire frequency ($P < 0.001$, standardized coefficient = 0.061) in western Canada, while the most important driving factors of Δ FFI in southern Europe and central China were mean cropland coverage ($P < 0.001$, standardized coefficient = 0.244) and cropland coverage change ($P < 0.001$, standardized coefficient = -0.132), respectively. For hotspots with increased FFI (Fig. 3d-f), wildfire frequency was the strongest driving factor of Δ FFI in the southeastern Amazon ($P < 0.001$, standardized coefficient = 0.299) and central Siberia ($P < 0.001$, standardized coefficient = 0.466), while Δ FFI in the Congo Basin

was significantly affected by all factors except nighttime light.”

Furthermore, a paragraph in the “Discussion” section was used to explain the effects of anthropogenic and natural factors on forest fragmentation dynamics in the revised version.

Line 308 to 324

“By coupling the changes in individual FFI components, we investigated Δ FFI and its associations with anthropogenic and natural factors and identified revealed the possible causes for forest fragmentation dynamics for some hotspots. For hotspots with increased FFI in the southeastern Amazon, large intact forest patches have been converted into multiple small patches under the mixed pressures from commercial harvest, cropland expansion and fire disturbances³⁹, causing serious forest losses and an increase in fragmentation (Supplementary Fig. 6). In central Siberia, however, forest losses due to fire disturbances, especially in forest edges, directly increased forest fragmentation during 2000-2020. For hotspots with decreased FFI in subtropical regions, especially in central China, the decrease in fragmentation was highly related to the implementation of ecological restoration projects under rapid economic development. For example, afforestation efforts under the “grain to green” project increased forest area and connected discrete forest patches⁴³. Also, for hotspots in western Canada and eastern Europe, the decline of FFI was mainly attributed to fire disturbances and changes in cropland area, respectively. These factors increase MPA, reduce ED and PD, and ultimately reduce forest fragmentation by smoothing forest edges and reducing small forest patches.”

[REFs 39 and 43 in revised manuscript]

39 Curtis, P. G., Slay, C. M., Harris, N. L., Tyukavina, A. & Hansen, M. C. Classifying drivers of global forest loss. *Science* 361, 1108-1111 (2018).

43 Lu, F. et al. Effects of national ecological restoration projects on carbon sequestration in China from 2001 to 2010. *Proc. Natl. Acad. Sci. U. S. A.* 115, 4039-4044 (2018).

Maybe my calculus is wrong, but I think there is a problem with the global forest cover extent calculations. The authors say (lines 384-389) that the $\approx 280,000 \times 25\text{km}^2$ grid cells contain almost all forest landscapes worldwide. However, a simple multiplication shows that it results in ≈ 7 million km^2 (700 million hectares) of forest cover worldwide, while the current FAO estimation for 2020 is about 4 billion hectares (FAO report).

Response: Thank you for your meticulous review of our manuscript. We apologize for the errors in the grid numbers of forest landscape in 2000 and 2020. According to the GLCLU forest map, the global forest area in 2020 was approximately 4.02 billion hectares (Potapov et al., 2022), which aligns with the value in the FAO report. Based on the updated forest cover data, the number of forest landscape grids in 2000 and 2020 are 3,413,077 and 3,422,375, respectively, with a total area of approximately 8.56 billion hectares for global forest landscapes in 2020. In our study, we defined a forest landscape as a 5km x 5km grid, with both forest and non-forest land cover types present in each landscape. Furthermore, the average forest coverage for global forest landscapes (with forest coverage > 30%) in 2020 was approximately 41.7%. Therefore, it is reasonable that the area of forest landscapes is larger than the actual forest area. We have revised the number of forest landscape grids and added further explanation in the revised manuscript.

Line 399 to 402

“In total 3,413,077 and 3,422,375 grid cells were considered for 2000 and 2020, respectively, which mainly covered all forest landscapes worldwide. The total area of forest landscapes was larger than the actual global forest area, because each forest landscape contains a certain proportion of non-forest areas.”

[REF]

Potapov, P. et al. The Global 2000-2020 Land Cover and Land Use Change Dataset Derived From the Landsat Archive: First Results. *Front. Remote Sens.* 3, 856903 (2022).

I understand that abbreviations can be practical, but the authors could reduce the number of acronyms within the text (or at least give the abbreviations full names within figure captions). More broadly, adding details on “how to interpret” the results in the figure captions would greatly help readers follow the text.

Response: Thank you for your suggestion. We have taken your suggestion and have made changes to our manuscript. We have reduced the number of abbreviations used and added more details about the results in the figure captions. For instance, we have replaced the abbreviations of FFMA (forest fragmentation mitigation areas) and FFEA (forest fragmentation exacerbation areas) with “areas with decreased FFI” and “areas with increased FFI” in the revised manuscript. Furthermore, we have included explanations about the meaning of “up” and “down” in the captions of Figures 2 and 4 to help readers better understand the contents.

Curtis et al. (2018, *Science*) have classified the main drivers of forest loss worldwide. The authors should cite and discuss their results with this previous work to link their results (trends in fragmentation dynamics) to drivers in different regions. They even could integrate the Curtis et al. (2018) results in their analyses.

Response: Thank you for your suggestion. We have carefully read the suggested paper (Curtis et al., 2018, *Science*) and found it very useful in explaining the causes of forest fragmentation change from 2000 to 2020 in some specific regions. We have cited this study in the revised manuscript.

Line 285 to 287

“In contrast, the land policy promulgated by the Brazilian government and its associated wildfire disturbances led to sharp forest losses³⁹ and significantly exacerbated forest fragmentation in the southeastern Amazon.”

Line 310 to 314

“For hotspots with increased FFI in the southeastern Amazon, large intact forest patches have been converted into multiple small patches under the mixed pressures from commercial harvest, cropland expansion and fire disturbances³⁹, causing serious forest losses and an increase in fragmentation (Supplementary Fig. 6).”

Furthermore, we have included an analysis of the driving forces behind forest fragmentation dynamics in our revised manuscript. The selection of explanatory variables was mainly based on Curtis et al.'s (2018) work.

Line 463 to 464

“Anthropogenic activities, natural disturbances, and socio-economic intensity are considered as the main drivers of global forest loss³⁹.”

[REF 39 in revised manuscript]

Curtis, P. G., Slay, C. M., Harris, N. L., Tyukavina, A. & Hansen, M. C. Classifying drivers of global forest loss. *Science* 361, 1108-1111 (2018).

I appreciate the figures, but I feel they could be better quality (e.g., vectorial).

Response: Thank you for your comment and suggestion regarding the quality of the figures. We have regenerated all of the figures using new forest cover data and uploaded them as individual high resolution TIF format files.

Reviewer #2 (Remarks to the Author):

I find the study to be a valuable approach to characterizing global forest change in the spatial domain. Examining forests fragmentation dynamically is the only way to go – we no longer need static maps. The approach is clear, although the shorthand (metric abbreviations) is a bit too dense to quickly uptake and interpret.

Response: We appreciate the time and effort you have dedicated to reviewing our manuscript. We have carefully revised the manuscript according to your suggestions and comments. Specifically, we have reduced the number of abbreviations in the manuscript. For example, we have replaced the abbreviations FFMA (forest fragmentation mitigation areas) and FFEA (forest fragmentation exacerbation areas) with the phrases "areas with decreased FFI" and "areas with increased FFI" respectively in the revised manuscript. In addition, we have added explanations about the meaning of "up" and "down" in the captions of Figures 2 and 4 to improve the manuscript's readability.

I find two major limitations of the study. First, the post-characterization comparison of two very different maps will result in incorrect change estimation. In remote sensing, the overlaying of two thematic maps contains true change and, of course, false change due to errors in either of the input maps (in most cases the two maps are made from similar data inputs, using the same method and the same definitions of themes of interest). Things get more problematic when comparing two very different thematic maps as the basis of comparison. The use of a 30m Landsat-based 2000 percent tree cover map compared to a discrete 10m Sentinel 2-based forest/non-forest map will have regional differences due to the varying methods, spatial scales, and nominal forest definitions used. When mapping forests, especially in areas where the formations are on the edge of the nominal definition, you will get false change. I believe boreal change in Central Russia could be one such example – a region where much of the landscape is on the edge of the physiognomic/structural definition of forest may yield different results in different mapping exercises. In any case, the definition of forest should be included – what is it in terms of height cover, minimum mapping unit? Are you able to truly harmonize the different maps, and if not, what is the impact or qualifications on the results? The paper presents all results as valid and unaffected by such impacts/considerations.

Response: Thank you for your valuable comments, which were also pointed out by the other two reviewers. We agree with you that the forest cover data from different data sources may introduce additional problems, besides errors in forest maps themselves, when comparing changes between two periods. We used the dataset from two very different maps because it was not possible to obtain forest extent maps for 2000 and 2020 at the very beginning of our study (**please see our responses**

to Reviewer 1). Following your suggestion, we changed to use forest extent maps for 2000 and 2020 from a study by Hansen's team on global land cover and land use (GLCLU) change during 2000-2020 (Potapov et al., 2022).

The study led by Hansen's team on global land cover and land use (GLCLU) change during 2000-2020 (Potapov et al., 2022) has released forest extent maps for 2000 and 2020. The GLULC forest data follows the definition of forest by the Food and Agriculture Organization of the United Nations (FAO), which sets "a pixel with tree height ≥ 5 m at the Landsat pixel scale as a forest pixel". However, in the previous manuscript, we used a threshold of 30% to convert the tree cover map into a forest map for 2000 and directly aggregated the 10 m resolution ESA forest cover (land cover) data into 30 m resolution forest maps for 2020. As you mentioned, these two forest maps actually have completely different definitions of forest pixels, and the comparison of the two forest maps, especially in regions with sparse forests, may result in significant errors. Fortunately, using forest extent maps from consistent datasets can reduce the uncertainties in forest cover comparisons caused by different forest definitions in various forest cover data sources, especially in areas where forest distribution is sparse (e.g. some boreal regions in the north part of Siberia and Far East Russia). Therefore, we used the two GLULC forest cover maps for 2000 and 2020 to replace the previous ones in the revised manuscript.

Based on the new forest cover data, we calculated the three components of forest fragmentation index (FFI), including edge density (ED), patch density (PD), and mean patch area (MPA) at 5km scale, regenerated the static and dynamic FFI distribution maps, and revised the results about the spatial patterns of the static and dynamic FFI. We also revised the contents about the modes of forest fragmentation processes and the assessment framework of forest landscape dynamics according to the new forest cover data.

Although there are some small differences of the magnitudes and distributions of forest fragmentation index (FFI) that calculated from current and previous data source versions, the main results and conclusions of our study do not change in general. The spatial distributions of the static FFI generally maintain the original patterns with the previous version. The intact forests (with low static FFI values) are mainly distributed in tropical regions, western Canada, and some areas in central Siberia and the Russian Far East, while high fragmented forests mainly located in eastern North America, South Europe, central and South China, and along the edges of tropical forest areas.

However, there are three major differences of the dynamic FFI (Δ FFI) between the current and previous results. First, the Δ FFI values in central Canada change from positive to negative. Second, the Δ FFI values in the edges of tropical forest of sub-Saharan Africa change from positive to negative. Third, the Δ FFI values in North Europe change from positive to negative. We simply compared Δ FFI of our results to Hansen's forest losses map (2001-2021) and found that the positive Δ FFI values has similar spatial distributions with forest losses regions. Although changes in FFIs and forest cover do not always keep in the same direction, we believe that the new forest cover dataset has somewhat eliminated some errors in the changes of FFIs due to different data sources, especially in areas such as southern Sweden and the northern region of tropical Africa.

All changes and revisions can be found in the revised version of manuscript.

[REF]

Potapov, P. et al. The Global 2000-2020 Land Cover and Land Use Change Dataset Derived From the Landsat Archive: First Results. *Front. Remote Sens.* 3, 856903 (2022).

The text describes findings with considerable qualified language regarding changes that “may be due to...” or “suggests that...” This is highly problematic as not all change may in fact be real. Some quantification of this issue is required, which comes to the second critique.

Attributes/drivers of forest fragmentation should be quantified, mainly through a sample-based assessment of the different categories at regional scales. The discussion of regional variations is too cursory and unsupported by data, when it could in fact be supported. If you were to examine a sample of each of your categories, you could verify which are definitive and which are not.

Response: We appreciate your insightful comments and agree that the lack of quantitative analysis regarding the driving forces of forest fragmentation dynamics in our previous manuscript rendered some of our conclusions unresponsive.

Therefore, in the revised manuscript, we conducted an analysis of the relationship between Δ FFI and various explanatory variables, including agricultural activity (mean cropland coverage and cropland coverage change), natural disturbance (fire frequency), socio-economic intensity (mean nighttime light and nighttime light change), and demographic pressure (mean population density and population density change), using general linear models. By doing so, we were able to identify the significance (p-value) and importance (standardized coefficient) of each factor in driving changes in forest fragmentation.

This driving force analysis was carried out for six hotspots (Fig. 3 in the revised version) and all 50 km size grids globally (Fig. S4 and S5 in the revised version). As a result, we have been able to provide more definitive expressions regarding the causes and processes of FFI changes in certain parts of the "Discussion" section.

Fig. 3 | Standardized correlation coefficients of the dynamic forest fragmentation index (Δ FFI) for

six hotspots. Relative effects of anthropogenic activity (the mean and difference values of cropland coverage and nighttime light, yellow color), demographic pressure (the mean and difference values of population density, blue color), and natural disturbance (fire frequency, red color) on dynamics of Δ FFI in (a)-(c) forest fragmentation decreased hotspots (D1-D3, n=7661, 3663, and 5271) and in (d)-(f) forest fragmentation increased hotspots (I1-I3, n=4044, 5104, and 7689). Dots represent standardized coefficient estimates with 95% (thin segments, ± 1.960 standard errors) and 90% (thick segments, ± 1.645 standard errors) confidence intervals in generalized linear models. Locations of the six hotspots can be checked in Fig. 2.

Fig. S4 | Standardized correlation coefficients of the dynamic forest fragmentation index (Δ FFI) for the globe. Relative effects of anthropogenic activity (the mean and difference values of cropland coverage and nighttime light; yellow color), demographic pressure (the mean and difference values of population density; blue color), and natural disturbance (fire frequency; red color) on dynamics of Δ FFI of all forested countries (n=131). Dots represented standardized coefficient estimates with 95% (thin segments, ± 1.960 standard errors) and 90% (thick segments, ± 1.645 standard errors) confidence intervals in generalized linear models.

Fig. S5 | Primary drivers of the dynamic forest fragmentation index (Δ FFI) for the period 2000 to 2020. The primary driver factor for each 50 km grid was represented by the factor with the highest absolute value of standardized coefficient estimates.

Results regarding the causes of forest fragmentation change during 2000-2020 can be checked in the section of “*Modes and causes of forest fragmentation processes*” in revised version.

Line 131 to 169

“We also detected the changes in individual fragmentation-related metrics for hotspots in areas with decreased and increased FFI, respectively. Overall, hotspots where FFI decreased had declines in ED and PD, and increases in MPA. Specifically, MPA increased significantly by 73% in western Canada, by 38% in southern Europe, and by 50% in central China from 2000 to 2020 (Fig. 2). Conversely, increased ED and PD played a more important role in hotspots where FFI increased. ED increased dramatically by 41% in the southeastern Amazon, by 81% in the Congo Basin, and by 90% in central Siberia, while the increments of PD in these hotspots were 32%, 186%, and 78%, respectively. However, MPA decreased slightly (-8% to -31%) in these hotspots where forest fragmentation increased.

Using generalized linear models, we further explored the relationships between Δ FFI and explanatory factors (see Methods) for the globe and the six hotspots. Although Δ FFI was not significantly correlated with any explanatory variables at the global scale (Supplementary Fig. 4), we found that anthropogenic activity factors (nighttime light, nighttime light change, cropland coverage, and cropland change) dominated the changes in FFI during 2000-2020 in the most developed areas, such as the eastern US, Europe, and South China (Supplementary Fig. 5). Moreover, wildfire mainly controlled the Δ FFI of some areas in Canada, Far East Russia, the southeastern Amazon, tropical Africa, and Australia. In addition, for hotspots with decreased FFI (Fig. 3a-c), Δ FFI was most strongly related to wildfire frequency ($P < 0.001$, standardized coefficient = 0.061) in western Canada, while the most important driving factors of Δ FFI in southern Europe and central China were mean cropland coverage ($P < 0.001$, standardized coefficient = 0.244) and cropland coverage change ($P < 0.001$, standardized coefficient = -0.132), respectively. For hotspots with increased FFI (Fig. 3d-f), wildfire frequency was the strongest driving factor of Δ FFI in the southeastern Amazon ($P < 0.001$, standardized coefficient = 0.299) and central Siberia ($P < 0.001$, standardized coefficient = 0.466), while Δ FFI in the Congo Basin was significantly affected by all factors except nighttime light.”

In addition, a paragraph in the “Discussion” section was used to explain the effects of anthropogenic and natural factors on forest fragmentation dynamics in the revised version.

Line 308 to 324

“By coupling the changes in individual FFI components, we investigated Δ FFI and its associations with anthropogenic and natural factors and identified revealed the possible causes for forest fragmentation dynamics for some hotspots. For hotspots with increased FFI in the southeastern Amazon, large intact forest patches have been converted into multiple small patches under the mixed pressures from commercial harvest, cropland expansion and fire disturbances ³⁹, causing serious forest losses and an increase in fragmentation (Supplementary Fig. 6). In central Siberia, however, forest losses due to fire disturbances, especially in forest edges, directly increased forest fragmentation during 2000-2020. For hotspots with decreased FFI in subtropical regions, especially in central China, the decrease in fragmentation was highly related to the implementation of ecological restoration projects under rapid economic development. For example, afforestation efforts under the “grain to green” project increased forest area and connected discrete forest patches ⁴³. Also, for hotspots in western Canada and eastern Europe, the decline of FFI was mainly attributed to fire disturbances and changes in cropland area, respectively. These factors increase MPA, reduce ED and PD, and ultimately reduce forest fragmentation by smoothing forest edges and reducing small forest patches.”

[REFs 39 and 43 in revised manuscript]

39 Curtis, P. G., Slay, C. M., Harris, N. L., Tyukavina, A. & Hansen, M. C. Classifying drivers of global forest loss. *Science* 361, 1108-1111 (2018).

43 Lu, F. et al. Effects of national ecological restoration projects on carbon sequestration in China from 2001 to 2010. *Proc. Natl. Acad. Sci. U. S. A.* 115, 4039-4044 (2018).

It is always incumbent on data producers to quantify uncertainties – where in the map are there results that are not supported by reference data, and likely a result of input data limitations? Along these lines, what might you omit or misinterpret using a bi-temporal comparison? Many regions have continuous dynamics, for example sub-tropical forestry, or humid tropical shifting cultivation. How to account for such regions in interpreting gain/loss dynamics?

Response: We appreciate for your valuable comments. We acknowledge that there are forest landscapes with continuous dynamics, particularly in tropical and subtropical regions. Nevertheless, due to the lack of global high-resolution forest cover data on an annual basis, it is still challenging to continuously monitor forest cover at a global scale, especially in tropical and subtropical regions. In the revised manuscript, we have addressed the uncertainties associated with detecting changes in forest fragmentation using only two-year forest cover data. We have also emphasized the need for continuous analysis of forest fragmentation to enhance its applications.

Line 339 to 345

“Moreover, it should be noted that the use of bi-temporal forest cover data also makes it impossible to fully assess the continuous dynamics, especially in some areas of sub-tropical forestry or humid tropical shifting cultivation. Therefore, more comprehensive analyses should be conducted that consider specific species characteristics, vegetation types, and multi-temporal forest cover data in the detection of the causes of forest fragmentation dynamics.”

Is mitigation the correct term if forest edge is lessened under a forestry land use regime? I am thinking of the Southeast USA, which I do not think has experienced any positive outcome regarding forest extent/arrangement in recent years, but is instead the site of highly intensive forestry land use. In general, the study, while a neat examination of dynamics, as it should be, does not look ‘under the hood’ enough to qualify or confirm findings.

Response: Thank you for your valuable comment. We agree that "mitigation" is not an appropriate term to describe the objective changes in forest fragmentation. Therefore, we have revised the text and replaced "mitigation" with "decrease" and "exacerbation" with "increase". However, we have used "mitigation" and "exacerbation" prudently in the Discussion section when the changes in FFIs and their causes are clear.

According to the new forest cover maps, both increased and decreased FFI areas can be found in the southeastern US. Coupling changes in FFI and FC, we further found that many areas with decreased FFI have a decreased FC ($FC_{\text{down}}FFI_{\text{down}}$) during 2000-2020. This is consistent with your views on the forest landscape changes in the region. Many forest landscapes in the southeastern US have degradation status over the past 20 years under the pressure of human activity (Fig. S5 in revised version).

Our results show that changes in FFI do not always consistent with the changes in FC. Increased FC may lead to increased FFI, while decreased FC may lead to decreased FFI. Therefore, the exploration of the causes of forest fragmentation needs to be combined with the dynamics of forest cover in specific locations, the modes of forest pattern change and the influences of related factors. In the revised manuscript, we have analyzed the relationship between ΔFFI and explanatory variables, including agricultural activity, natural disturbance, socio-economic intensity, and demographic pressure using general linear models. We identified the significance (p-value) and importance (standardized coefficient) of each factor on forest fragmentation dynamics using the general linear models. This driving force analysis was conducted for 6 hotspots (Fig. 3 in revised version) and all 50 km size grids in the globe (Fig. S4 and S5 in revised version). We expect to reveal the possible causes of FFI dynamics through the driving force analysis of forest fragmentation in the 50 km grids of the globe and hotspots. However, this analysis may not be able to explain FFI dynamics in all regions. We mentioned these uncertainties in the discussion in the revised manuscript.

All mentioned figures of Fig. 3, Fig. S4, and Fig. S5 have been shown in above responses.

Line 334 to 347

“However, the complexities of forest fragmentation processes and the causes also remind us that forest fragmentation studies should be targeted and localized. The availability of precise forest distribution data and a thorough understanding of forest landscape dynamic drivers, including land policy, climate change, and international trade, are essential conditions for research on the patterns, causes, ecological consequences, and coping strategies of forest fragmentation. Moreover, it should be noted that the use of bi-temporal forest cover data also makes it impossible to fully assess the continuous dynamics, especially in some areas of sub-tropical forestry or humid tropical shifting cultivation. Therefore, more comprehensive analyses should be conducted that consider specific species characteristics, vegetation types, and multi-temporal forest cover data in the detection of the causes of forest fragmentation dynamics. These analyses also form the basis for applying the assessment of patterns and causes of forest fragmentation to biodiversity conservation and carbon

cycle feedback mechanisms.”

A couple of minor queries.

Why 5000km²? Did you assess various scales of aggregation or was this simply the most computationally doable resolution? What impact does the choice of output grid cell size have on results?

Response: Thank you for your comments. Forest fragmentation is a landscape-level phenomenon and reflects the dynamics of forest distribution pattern. Therefore, we quantified forest fragmentation and its change at landscape scale in our study. Considering that landscape is usually defined as an area of tens to hundreds of square kilometers. We initially carried out the calculation of forest landscape pattern indexes at two scales (5km size grid and 25km size grid). In addition, the limitation of computing power is also a reason that we did not conduct FFI calculations at more spatial scales. It takes about 10 days to complete the calculation of three landscape pattern metrics using 5000 m size grids and 30 m resolution forest extent map of one year.

Based on the results, we found that the spatial distributions of the three individual landscape metrics of different spatial scales basically maintained the same pattern (Fig. R1). Therefore, we believe that the grid size may not have a significant effect on the forest fragmentation pattern. However, we acknowledge that the spatial resolution may affect the estimation of certain fragmentation metrics and our ability to detect small-scale fragmentation processes.

Given the importance of providing a high resolution of forest fragmentation results, we chose to use the 5km size grid to display the final results. We believe that this scale is a good compromise between providing a high resolution of results and minimizing the impact of computational limitations.

Fig. R1 | Spatial distributions of the three individual landscape metrics (edge density, patch density, and mean patch area) of the year 2020 at different spatial scales (different size grids). The high (5 km) and low (25 km) spatial resolutions of landscape metrics are shown respectively in left and

right column.

Also, please explain more clearly to readers how fragmentation can change without forest loss or gain. I am thinking of the line where FFI “increased in central Europe between 2000 and 2020, in which no remarkable forest cover change (loss or gain) was detected.” You should show some examples of the various dynamics at full resolution, meaning native map input resolutions, to facilitate understanding of your coarser scale metrics, something along the lines of a tutorial. You could really help users get from the inputs to the metrics through some detailed examples.

Response: Thank you for your questions and suggestions. As to the meaning of the sentence “For example, our results showed that the FFI decreased in western Canada and the central Amazon but **increased in central Europe between 2000 and 2020, in which no remarkable forest cover change (loss or gain) was detected**”, we wanted to show that, according to previous FFI assessments, FFI increased in central Europe during 2000-2020, but according to other studies (Hansen et al., 2013; Tyukavina et al., 2022), there is no significant forest loss in this region.

In the revised manuscript, the FFIs for 2000 and 2020, derived from new forest cover data, have some changes when compared with the previous version. We have deleted the original content and replaced it with the new content to show that the changes in FFIs is not always consistent with the change in FC.

Line 248 to 253

“For example, we found that the FFI decreased in some areas in central Canada and the central Amazon between 2000 and 2020, but forest losses were still found in those areas. These findings further emphasize the value of our proposed two-dimensional assessment framework in the evaluation of forest landscape pattern dynamics, and it also provides reasonable approaches for evaluating forest fragmentation and its dynamics at regional-national-global scales.”

In response to your suggestions, we have incorporated a high-resolution forest cover map for each hotspot, represented by a 5 km × 5 km size grid, in order to assist readers and users in comprehending the connection between forest cover, forest landscape metrics, and FFI (Fig. S6 in the revised version). This figure provides valuable information regarding the analysis of the causes of forest fragmentation, as it directly identifies changes in forest cover map and FFI, including its components, in specific areas. For instance, in the Amazon (II) pixel, we can observe the conversion of intact forest landscapes into fragmented forest patches, with the remaining forest patches characterized by relatively neat edges. This indicates that forest harvest was the primary cause of fragmentation in numerous areas of the Amazon.

[REFs]

Hansen, M. C. et al. High-resolution global maps of 21st-century forest cover change. *Science* 342, 850-853 (2013).

Tyukavina, A. et al. Global Trends of Forest Loss Due to Fire From 2001 to 2019. *Front. Remote Sens* 3, 825190 (2022).

Fig. S6 | Spatial distribution of full resolution of forest cover change from 2000 to 2020 for some sites of different locations of the globe. Each of the site is a 5 km size grid and was selected in relevant hotspots in forest fragmentation decreased and increased areas of Fig. 2.

Reviewer #3 (Remarks to the Author):

General comments

This study analyzed the changes in the forest fragmentation index in a period to obtain the dynamic fragmentation index and its global pattern. It quantified global patterns of forest fragmentation from a perspective more consistent with the concept of forest fragmentation that cannot be reflected by the static fragmentation index. By analyzing the combination of changes in forest fragmentation and forest coverage, this study developed a framework to quantify the spatial distribution of global forest landscape dynamic patterns (degradation or restoration), discussed the important ecological consequences of forest fragmentation, and provide possible mitigation approaches. This study tries to examine forest fragmentation from a dynamic perspective (fragmentation itself has the nature of change) and prove through quantitative analysis that dynamic FFI can better reflect forest landscape changes in many hot spots around the world than static FFI. I believe this study has important merit in the field of forest landscape dynamic evaluation, and the related forest fragmentation and forest landscape dynamic maps are potentially valuable to stakeholders in both basic research and policy-oriented roles. The article is well-written and the logic is clear. However, I have several major points I think should be considered before its consideration for publication.

Response: Thank you for your valuable and constructive comments. We have thoroughly reviewed your feedback and made every effort to address each point in our revision. We hope the revision can meet with approval.

1) The quality, consistency in definition and classification approaches, and the accuracies of two baseline forest maps are key for the reliability of the results. How can the authors confirm that the forest information from the Hansen forest map and ESA WorldCover map can be compared directly? Why are the results not the artifacts of the biased forest maps-based analyses?

Response: We appreciate for your valuable comments. It is really true as you mentioned that data sources have significant impacts on the detection of changes in forest fragmentation index (FFI). We should not directly compare forest maps of different data sources. In order to exclude the systematic errors induced by forest cover data source, we have used another forest extent product (Global Land use and land cover change) in the revision.

At the very beginning of study (June 2021), we planned to use multiple years of forest cover data that derived from Hansen's tree cover map. However, we can only find tree cover map for 2000, forest losses layers for 2001-2021, and forest gains layers for 2001-2012 in the datasets (include updated datasets) related to Hansen's Science paper (Global Land Analysis & Discovery, <https://glad.umd.edu/dataset>). Based on these data, we cannot obtain the tree cover map for 2020. We also emailed to Dr. Matthew Hansen to ask if they have tree cover data for 2020 or adjacent years (using the same method as the tree cover map of 2000 in their Science paper), but we didn't get a positive answer. Actually, we also tried to obtain consistent forest cover maps for 2000 and 2020 from other data source. We analyzed forest distribution maps derived from tree cover maps of Global Forest Cover Change (GFCC) dataset. But the GFCC tree cover dataset that publicly available only covers the period between 2000 and 2015. We noticed the GFCC tree cover map in 2020 may be available from a commercial website (www.terraPulse.com). We sent messages on the website about how to get the GFCC tree cover data in 2020, but we got no response. Considering that the global forest cover and forest landscape pattern may have large changes during 2015-2020, especially in tropical areas (e.g. Brazilian Amazon), we decided to use forest cover data in 2020

derived from ESA land cover map instead, and to compare it with forest cover data in 2000 derived from Hansen's tree cover map in our forest fragmentation change analysis.

After read and think of the comments from three reviewers' comments carefully, we agree that using forest cover maps from different data sources are more problematic compared to those from the same data source. Luckily, we noticed another study by Hansen's team on global land cover and land use (GLCLU) change during 2000-2020 (Potapov et al., 2022), which released forest extent maps for 2000 and 2020. The definition of forest set as "a pixel with tree height ≥ 5 m at the Landsat pixel scale" in GLULC forest maps, which is consistent with the definition by the Food and Agriculture Organization of the United Nations (FAO). Therefore, we used the two forest cover maps for 2000 and 2020 to replace the previous ones.

Based on the new forest cover data, we calculated the three components of forest fragmentation index (FFI), including edge density (ED), patch density (PD), and mean patch area (MPA) at 5km scale, regenerated the static and dynamic FFI distribution maps, and revised the results about the spatial patterns of forest fragmentation and its change. We also revised the contents about the modes of forest fragmentation processes and the assessment framework of forest landscape dynamics according to the new forest cover data.

Although there are some small differences of the magnitudes and distributions of forest fragmentation index (FFI) after using the updated datasets, the main results and conclusions of our study do not change. All changes and revisions can be found in the revised manuscript.

[REF]

Potapov, P. et al. The Global 2000-2020 Land Cover and Land Use Change Dataset Derived From the Landsat Archive: First Results. *Front. Remote Sens.* 3, 856903 (2022).

2) Three landscape pattern indexes (edge density, patch density, and mean patch area) were considered for the construction of the integrated forest fragmentation index (FFI). According to the global maps of the three indexes from the supplementary information, we found these three individual landscape metrics have similar spatial patterns. So, it is necessary to use three indexes to construct FFI?

Response: Thank you for your questions. The conversion of intact forest landscapes into fragmented states results in a range of distinct characteristics, including increased edge length and edge density, an increase in the number of forest patches, and a decrease in the mean area of forest patches. These characteristics respectively represent the edge effect, isolation effect, and patch area effect of forest fragmentation (Haddad et al., 2015). Such features demonstrate the impact of forest fragmentation on ecosystem processes and functions, such as carbon sequestration, microclimate regulation, and biodiversity maintenance. Although different landscape pattern indexes can be utilized to describe fragmentation, each index reflects a specific aspect of the nature of forest fragmentation within the same landscape. Hence, it is reasonable to expect that different landscape pattern indexes exhibit similar spatial patterns in general. Nevertheless, there remain numerous nuanced differences among the spatial distributions of various forest landscape indexes. This is a crucial reason why we aimed to incorporate a variety of forest landscape indexes in our study to quantify the comprehensive characteristics of forest fragmentation.

[REF]

Haddad, N. M. et al. Habitat fragmentation and its lasting impact on Earth's ecosystems. *Sci. Adv.* 1, e1500052 (2015).

If so, a more detailed reason for the selection of the three metrics should be provided. In addition, why did the authors use equal weights (if I understand correctly) of the three metrics in calculating FFI?

Response: Thank you for your comments and suggestions. We have revised the sentence that describing the reason of the using of three landscape indexes (ED, PD, and MPA).

Line 404 to 408

“Edge effect, isolation effect, and patch size effect were the most important features of forest fragmentation⁵, and they can be quantified by three landscape pattern metrics, including edge density (ED), patch density (PD) and mean patch area (MPA), respectively. The three landscape pattern metrics were used to assemble a synthetic forest fragmentation index in our study.”

[REF 5 in the revised manuscript]

5 Haddad, N. M. et al. Habitat fragmentation and its lasting impact on Earth's ecosystems. *Sci. Adv.* 1, e1500052 (2015).

Regarding the weights assigned to each landscape index, we evaluated forest fragmentation from a spatial pattern perspective rather than considering specific features and their ecological effects. Therefore, it was reasonable to assign equal weights to each landscape index in our study. In addition, the spatial distribution maps of individual landscape pattern indexes (Fig. S1 in revised version) were also shown in the supplementary materials, and these layers may play an important role in studying the effects of specific aspect of forest fragmentation.

Fig. S1 | The spatial distribution of (a-b) edge density (ED), (c-d) mean patch area (MPA), and (e-f) patch density (PD) for global forest landscapes in 2000 and 2020. The range of values for each

landscape pattern index is 0-1. More details of the specific calculation formulas were mentioned in the Methods section of the main text and Extended methods section of the Supplementary Information.

3) The attribution of the global forest fragmentation changes were kind of weak, and lack of quantitative analyses. I would suggest the authors build a model to consider a few natural and socioeconomic drivers (e.g., climate change, population, cropland expansion, plantations, etc.) in different countries.

Response: We appreciate for your comments and suggestions. We recognize the importance of exploring the reasons and causes of forest fragmentation changes, and have incorporated this aspect into our study. Specifically, we analyzed the changes in FFI components (i.e., edge density, patch density, and mean patch area) from 2000 to 2020 in six hotspots. These hotspots were selected based on regions with significant FFI changes, with three in the FFI increased area and three in the FFI decreased area (Fig. 2 in the revised version). We also investigated the relationship between Δ FFI and various explanatory factors, such as agricultural activity (mean cropland coverage and cropland coverage change), natural disturbance (fire frequency), socio-economic intensity (mean nighttime light and nighttime light change), and demographic pressure (mean population density and population density change), and determined the significance (p-value) and importance (standardized coefficient) of each factor using a general linear model.

Initially, we included climate change factors (i.e., trends of maximum temperature, minimum temperature, and precipitation from 2000 to 2020) in our analysis. However, we found that the relationship between climate change factors and forest cover change was extremely weak. Moreover, previous studies (Curtis et al., 2018) have not identified climate factors as major drivers of global forest losses. Therefore, we excluded climate change factors from our explanatory variable list.

This driving force analysis was conducted for 6 hotspots (Fig. 3 in revised version) and all 50 km size grids in the globe (Fig. S4 and S5 in revised version).

Results regarding the causes of forest fragmentation change during 2000-2020 can be checked in the section of “*Modes and causes of forest fragmentation processes*” in revised version.

Line 131 to 169

“We also detected the changes in individual fragmentation-related metrics for hotspots in areas with decreased and increased FFI, respectively. Overall, hotspots where FFI decreased had declines in ED and PD, and increases in MPA. Specifically, MPA increased significantly by 73% in western Canada, by 38% in southern Europe, and by 50% in central China from 2000 to 2020 (Fig. 2). Conversely, increased ED and PD played a more important role in hotspots where FFI increased. ED increased dramatically by 41% in the southeastern Amazon, by 81% in the Congo Basin, and by 90% in central Siberia, while the increments of PD in these hotspots were 32%, 186%, and 78%, respectively. However, MPA decreased slightly (-8% to -31%) in these hotspots where forest fragmentation increased.”

Using generalized linear models, we further explored the relationships between Δ FFI and explanatory factors (see Methods) for the globe and the six hotspots. Although Δ FFI was not significantly correlated with any explanatory variables at the global scale (Supplementary Fig. 4), we found that anthropogenic activity factors (nighttime light, nighttime light change, cropland coverage, and cropland change) dominated the changes in FFI during 2000-2020 in the most

developed areas, such as the eastern US, Europe, and South China (Supplementary Fig. 5). Moreover, wildfire mainly controlled the Δ FFI of some areas in Canada, Far East Russia, the southeastern Amazon, tropical Africa, and Australia. In addition, for hotspots with decreased FFI (Fig. 3a-c), Δ FFI was most strongly related to wildfire frequency ($P < 0.001$, standardized coefficient = 0.061) in western Canada, while the most important driving factors of Δ FFI in southern Europe and central China were mean cropland coverage ($P < 0.001$, standardized coefficient = 0.244) and cropland coverage change ($P < 0.001$, standardized coefficient = -0.132), respectively. For hotspots with increased FFI (Fig. 3d-f), wildfire frequency was the strongest driving factor of Δ FFI in the southeastern Amazon ($P < 0.001$, standardized coefficient = 0.299) and central Siberia ($P < 0.001$, standardized coefficient = 0.466), while Δ FFI in the Congo Basin was significantly affected by all factors except nighttime light.”

In addition, a paragraph in the “Discussion” section was used to explain the effects of anthropogenic and natural factors on forest fragmentation dynamics in the revised version.

Line 308 to 324

“By coupling the changes in individual FFI components, we investigated Δ FFI and its associations with anthropogenic and natural factors and identified revealed the possible causes for forest fragmentation dynamics for some hotspots. For hotspots with increased FFI in the southeastern Amazon, large intact forest patches have been converted into multiple small patches under the mixed pressures from commercial harvest, cropland expansion and fire disturbances³⁹, causing serious forest losses and an increase in fragmentation (Supplementary Fig. 6). In central Siberia, however, forest losses due to fire disturbances, especially in forest edges, directly increased forest fragmentation during 2000-2020. For hotspots with decreased FFI in subtropical regions, especially in central China, the decrease in fragmentation was highly related to the implementation of ecological restoration projects under rapid economic development. For example, afforestation efforts under the “grain to green” project increased forest area and connected discrete forest patches⁴³. Also, for hotspots in western Canada and eastern Europe, the decline of FFI was mainly attributed to fire disturbances and changes in cropland area, respectively. These factors increase MPA, reduce ED and PD, and ultimately reduce forest fragmentation by smoothing forest edges and reducing small forest patches.”

[REFs 39 and 43 in revised manuscript]

39 Curtis, P. G., Slay, C. M., Harris, N. L., Tyukavina, A. & Hansen, M. C. Classifying drivers of global forest loss. *Science* 361, 1108-1111 (2018).

43 Lu, F. et al. Effects of national ecological restoration projects on carbon sequestration in China from 2001 to 2010. *Proc. Natl. Acad. Sci. U. S. A.* 115, 4039-4044 (2018).

Fig. 2 | Spatial distributions and composition proportions of eight forest fragmentation process modes for forest fragmentation decreased areas (the upper row) and forest fragmentation increased areas (the below row). ED, PD, and MPA mean the individual components of the synthetic forest fragmentation index (FFI). The marks of “up” and “down” after each FFI component represent an increase and decrease trend during 2000-2020, respectively. Three hotspots in the most obvious FFI decreased and increased areas were selected respectively to evaluate the changes in each component of FFI. Hotspots D1-D3: western Canada, southern Europe, and central China, and hotspots I4-I6: southeastern Amazon, Congo Basin, and central Siberia.

Fig. 3 | Standardized correlation coefficients of the dynamic forest fragmentation index (Δ FFI) for six hotspots. Relative effects of anthropogenic activity (the mean and difference values of cropland coverage and nighttime light, yellow color), demographic pressure (the mean and difference values of population density, blue color), and natural disturbance (fire frequency, red color) on dynamics of Δ FFI in (a)-(c) forest fragmentation decreased hotspots (D1-D3, n=7661, 3663, and 5271) and in (d)-(f) forest fragmentation increased hotspots (I1-I3, n=4044, 5104, and 7689). Dots represent standardized coefficient estimates with 95% (thin segments, ± 1.960 standard errors) and 90% (thick segments, ± 1.645 standard errors) confidence intervals in generalized linear models. Locations of the six hotspots can be checked in Fig. 2.

Fig. S4 | Standardized correlation coefficients of the dynamic forest fragmentation index (Δ FFI) for the globe. Relative effects of anthropogenic activity (the mean and difference values of cropland coverage and nighttime light; yellow color), demographic pressure (the mean and difference values of population density; blue color), and natural disturbance (fire frequency; red color) on dynamics of Δ FFI of all forested countries (n=131). Dots represented standardized coefficient estimates with 95% (thin segments, ± 1.960 standard errors) and 90% (thick segments, ± 1.645 standard errors) confidence intervals in generalized linear models.

Fig. S5 | Primary drivers of the dynamic forest fragmentation index (ΔFFI) for the period 2000 to 2020. The primary driver factor for each 50 km grid was represented by the factor with the highest absolute value of standardized coefficient estimates.

Also, percolation theory has been considered in the global patterns of tropical forest fragmentation (Taubert et al., Nature 2018), Has this study considered those influence factors in the previous study?

Response: Thank you for your comments. We have taken note of the paper you mentioned and appreciate the insights it offers regarding the relationship between patch number and patch area of tropical forest patches. However, it is important to note that our study differs from the aforementioned paper in its focus on assessing global forest fragmentation and its changes at the landscape scale through the characterization of its spatial distribution. While percolation theory may be well-suited for explaining the relationship between patch number and patch area in certain contexts, its usefulness in quantifying the spatial distribution characteristics of forest fragmentation is limited. Thus, we have not applied percolation theory in our assessment of forest fragmentation and its changes.

4) In the analysis of forest fragmentation process modes obtained based on the changes of three individual fragmentation-related landscape metrics, the possible meanings represented by atypical modes (modes except A for forest fragmentation exacerbation area and modes except for H for forest fragmentation mitigation area) are not discussed enough.

Response: Thank you for your comments. We have incorporated your feedback and have revised the manuscript accordingly. Specifically, we have added a section to discuss the significance of atypical modes in reflecting forest fragmentation processes and causes.

Line 301 to 307

“However, we also found some atypical FFI change modes. For example, in the central Amazon, MPA decreased in some areas where FFI decreased, while PD decreased in some areas where FFI decreased, which indicated that the processes of forest fragmentation were extremely complex. Therefore, efforts in detecting the underlying mechanism of forest fragmentation change should be site-specific, and focus on the relationship between explanatory factors and forest landscape patterns.”

Furthermore, the reasons for the selection of the six hotspots during the analysis of the fragmentation

process pattern need to be strengthened. In addition, the authors can think about moving Fig S3 into the main text given the truth that it can demonstrate changes in various metrics in different fragmentation processes.

Response: Thank you for your suggestions. We have incorporated your feedback by moving Fig. S3 into the main text and merging it with Fig. 2, which can now be found in the revised manuscript as Fig. 2. Regarding the selection of hotspots in this study, we based our choices mainly on the locations where the most significant changes in forest fragmentation occurred during the period of 2000-2020. We identified major regions that exhibited substantial changes in forest fragmentation and selected three hotspots in areas where fragmentation had decreased markedly and three hotspots in areas where fragmentation had increased markedly.

Line 453 to 456

“To better analyze the processes driving changes in forest fragmentation, we selected three hotspots (western Canada, southern Europe, and central China) where fragmentation was remarkably decreased and three hotspots (the southeastern Amazon, the Congo Basin, and central Siberia) where fragmentation was obviously increased.”

Specific comments

Line 45: What do you mean by the “first several decades”? The community would agree that the most recent few decades are the best stage for understanding forest fragmentation due to the availability of remote sensing-based forest maps.

Response: Thank you for your question. The meaning of the term “first several decades” refers that the ecological effects of forest fragmentation are more obvious in the first few decades after the formation of the fragmentation pattern. The impacts of forest fragmentation on ecosystem structure and function are gradually decreased as time goes by. Selecting a suitable time range to examine the change of forest fragmentation is an important basis to apply the results of forest fragmentation assessment. We believe 20 years is a appropriate time range to reflect the features of forest fragmentation change. Considering the availability of remote sensing data, we chose 2000-2020 as the research period in our study. We have revised this sentence to help readers understand the meaning of the sentence accurately.

Line 44 to 46

“Moreover, although the effects of forest fragmentation can persist for up to a century¹⁶, the effects are mostly immediate and obvious for only the first several decades after formation^{17,18}.”

[REFs 16-18 in revised manuscript]

16 Vellend, M. et al. Extinction debt of forest plants persists for more than a century following habitat fragmentation. *Ecology* 87, 542-548 (2006).

17 Laurance, W. F. et al. Ecosystem decay of Amazonian forest fragments: A 22-year investigation. *Conserv. Biol.* 16, 605-618 (2002).

18 Gibson, L. et al. Near-Complete Extinction of Native Small Mammal Fauna 25 Years After Forest Fragmentation. *Science* 341, 1508-1510 (2013).

Line 80: There might be an error here. Change 0.2 to 0.8?

Response: Thank you for your comment, and we have revised this error.

Figure 3a: in the legend, “FC” can be moved ahead of “FFI”

Response: Amended as required. Thank you.

Line 387: why are the grids in the two years different?

Response: Thank you for your question. We generated the grids according to the forest distribution in two years, respectively. Since the forest distribution and forest area of the two years are different, the number of grids in the two years will also be different. We only used the overlaid parts of the grids of the two years to analyze the changes in forest fragmentation.

REVIEWERS' COMMENTS

Reviewer #1 (Remarks to the Author):

The authors have done much work to address my comments (and those given by other reviewers), including further in-depth analyses with a much more reliable dataset. I believe the manuscript is now ready for publication.

Reviewer #2 (Remarks to the Author):

I am satisfied with the revision, particularly the consistent data set used to perform the analysis.

Reviewer #3 (Remarks to the Author):

The authors have addressed all of my concerns and questions from the previous round of review. Especially they changed the data source of forest cover and added analysis of the driving forces of forest fragmentation in the revised version, making the results more robust. I believe the authors have done a good job in improving the study.

I only have one more minor comment:

Line 463-464, change the "socio-economic intensity" to "demographic pressure." I think the authors are trying to divide explanatory variables into three categories (anthropogenic activity, demographic pressure, and natural disturbance), and socio-economic intensity should belong to the anthropogenic activity category.

Response to Reviewer's Comments

Reviewer #1 (Remarks to the Author):

The authors have done much work to address my comments (and those given by other reviewers), including further in-depth analyses with a much more reliable dataset. I believe the manuscript is now ready for publication.

Response: Thank you very much for your time and effort in reviewing our manuscript. Your comments and suggestions have played a very important role in improving our study.

Reviewer #2 (Remarks to the Author):

I am satisfied with the revision, particularly the consistent data set used to perform the analysis.

Response: We appreciate for your review of our manuscript. Your comments and suggestions are important for enhancing our study.

Reviewer #3 (Remarks to the Author):

The authors have addressed all of my concerns and questions from the previous round of review. Especially they changed the data source of forest cover and added analysis of the driving forces of forest fragmentation in the revised version, making the results more robust. I believe the authors have done a good job in improving the study.

Response: Thank you very much for your positive comments.

I only have one more minor comment:

Line 463-464, change the "socio-economic intensity" to "demographic pressure." I think the authors are trying to divide explanatory variables into three categories (anthropogenic activity, demographic pressure, and natural disturbance), and socio-economic intensity should belong to the anthropogenic activity category.

Response: Thank you for your comment. We have changed "socio-economic intensity" into "demographic pressure" in this sentence.

Line 433 to 434

"Anthropogenic activities, demographic pressure, and natural disturbances are considered as the main drivers of global forest loss."